# Dysfunctional ERG signaling drives pulmonary vascular aging and persistent fibrosis

Nunzia Caporarello [1,7], Jisu Lee[2,7], Tho X. Pham[2], Dakota L. Jones[3], Jiazhen Guan[2], Patrick A. Link [1], Jeffrey A. Meridew[1], Grace Marden[2], Takashi Yamashita [2], Collin A. Osborne[4], Aditya V. Bhagwate[4], Steven K. Huang[5], Roberto F. Nicosia[6], Daniel J. Tschumperlin[1], Maria Trojanowska [2] & Giovanni Ligresti [2✉]

Vascular dysfunction is a hallmark of chronic diseases in elderly. The contribution of the vasculature to lung repair and fibrosis is not fully understood. Here, we performed an epigenetic and transcriptional analysis of lung endothelial cells (ECs) from young and aged mice during the resolution or progression of bleomycin-induced lung fibrosis. We identified the transcription factor ETS-related gene (ERG) as putative orchestrator of lung capillary homeostasis and repair, and whose function is dysregulated in aging. ERG dysregulation is associated with reduced chromatin accessibility and maladaptive transcriptional responses to injury. Loss of endothelial ERG enhances paracrine fibroblast activation in vitro, and impairs lung fibrosis resolution in young mice in vivo. scRNA-seq of ERG deficient mouse lungs reveals transcriptional and fibrogenic abnormalities resembling those associated with aging and human lung fibrosis, including reduced number of general capillary (gCap) ECs. Our findings demonstrate that lung endothelial chromatin remodeling deteriorates with aging leading to abnormal transcription, vascular dysrepair, and persistent fibrosis following injury.

[1] Department of Physiology & Biomedical Engineering, Mayo Clinic, Rochester, MN, USA. [2] Department of Medicine, Boston University School of Medicine, Boston, MA, USA. [3] Department of Medicine, Perelman School of Medicine, University of Pennsylvania, Philadelphia, PA, USA. [4] Department of Biomedical Statistics and Informatics, Mayo Clinic, Rochester, MN, USA. [5] Department of Internal Medicine, University of Michigan Medical School, Ann Arbor, MI, USA. [6] Department of Laboratory Medicine and Pathology, University of Washington, Seattle, WA, USA. [7] These authors contributed equally: Nunzia Caporarello, Jisu Lee. ✉email: ligresti@bu.edu

Endothelial cells (ECs) exhibit remarkable heterogeneity and plasticity to adapt to different tissue requirements under physiological and pathological conditions[1]. ECs are responsible for numerous functions, including organ development, regeneration, and repair[2]. These critical functions, however, often decline with aging leading to abnormal homeostasis and dysrepair following tissue injury[3,4]. Since vascular regeneration is critical to tissue repair[5], its failure may contribute to the development of aging-related disorders, including chronic lung diseases[6,7]. Among these, idiopathic pulmonary fibrosis (IPF) is a progressive and fatal aging-related disease[8,9] characterized by aberrant wound healing, exaggerated extracellular matrix (ECM) deposition[10,11], and vascular alterations[12–14]. Notably, a recent study correlated lung vascular rarefaction with an increased risk of developing interstitial lung disease[15]. Nevertheless, most studies in IPF have focused on epithelial injury and fibroblast responses, and the role of the vascular endothelium in the pathogenesis of this disease has remained largely unexplored.

Using a mouse model of bleomycin-induced lung fibrosis we demonstrated that aging promotes persistent lung fibrosis associated with dysfunctional endothelial transcriptional responses and abnormal vascular remodeling[16]. Recent studies have revealed that lung ECs are transcriptionally heterogeneous[17–19]. One of these studies identified two functionally distinct populations of capillary ECs in the alveolar septum: the aCap ECs (also termed aerocytes), which specialize in gas exchange, and the gCap ECs (general capillary), which function as progenitor cells that give rise to and replenish alveolar-capillary ECs during maintenance and repair after injury[17]. Although these studies suggest that pulmonary ECs activate specific cellular and transcriptional programs to promote vascular repair, the transcriptional and epigenetic mechanisms governing endothelial responses during aging and persistent lung fibrosis remain unexplored.

Here, we used a comparative epigenetic (assay for transposase-accessible chromatin using sequencing— ATAC-seq) and transcriptional (RNA sequencing-RNA-seq) analysis on freshly sorted mouse lung ECs and discovered that young pulmonary vasculature responded to bleomycin injury by increasing chromatin accessibility and by activating transcriptional programs leading to vascular repair and fibrosis resolution. Aged lung endothelium, however, failed to activate these programs, resulting in vascular dysrepair and sustained fibrosis. We identified the endothelial transcription factor ETS-related gene ERG as a putative orchestrator of endothelial chromatin remodeling following lung injury, whose function decreases in aging. ERG was previously implicated in regulating endothelial lineage specification, vascular homeostasis, and angiogenesis, and its dysregulation led to increased inflammation and aberrant tissue remodeling in the liver and heart[20–22]. Here, we demonstrated that ECs lacking ERG enhanced fibrogenic responses in lung fibroblasts in vitro and perpetuated lung fibrosis in vivo. Whole lung scRNA-seq on endothelial ERG-deficient mice revealed a reduced number of gCap ECs and a dysfunctional transcriptional signature mirroring that of aged ECs. Interrogation of scRNA-seq datasets from human IPF and healthy lungs[23] also revealed a reduced number of gCap ECs and limited expression of gCap EC specific markers in IPF lungs compared to healthy ones, thereby phenocopying the endothelial responses of the fibrotic aged mouse lungs.

Taken together, these findings suggest that ERG-mediated chromatin remodeling governs cellular and transcriptional programs to promote vascular repair following injury and that aberrant ERG/chromatin dynamics in aged lung ECs leads to vascular dysrepair and fibrosis.

## Results
### Landscape of chromatin accessibility in lung ECs from young and aged mice.
We and others have previously reported that in young mice lung fibrosis resolves following a single dose of bleomycin[8,24,25]. The pulmonary vasculature in young mice responds to bleomycin-induced lung injury by upregulating numerous genes implicated in vascular remodeling[16]. Conversely, in the lungs of aged mice, numerous endothelial genes implicated in angiogenesis fail to be upregulated, suggesting that aging attenuates endothelial transcriptional responses to lung injury[16].

Given that chromatin remodeling is essential to coordinate gene transcription[26], we carried out a chromatin analysis using ATAC-seq on lung ECs isolated from young and aged mice at baseline and during the early resolution phase of bleomycin-induced lung fibrosis (30 days post-bleomycin) (Supplementary Fig. 1). Principal component analysis (PCA) revealed differences in chromatin accessibility, with samples clustering in two distinct groups based on age (PCA-2) (Fig. 1a). Chromatin peaks in lung ECs from aged mice (Sham and Bleomycin-treated) were overall lower than those of ECs from young lungs, demonstrating a widespread decrease of chromatin accessibility with aging (Fig. 1b and Supplementary Fig. 2). We then performed the following comparative analysis: uninjured aged ECs relative to uninjured young ECs (Aged vs Young), injured young ECs relative to uninjured young ECs (Bleomycin response–Young) and injured aged ECs relative to uninjured young ECs (Bleomycin response–Aged). This analysis confirmed the limited chromatin accessibility in lung ECs from aged mice in absence of injury (Fig. 1c). Interestingly, the lung endothelial chromatin of injured young mice showed increased accessibility whereas that of injured aged lung ECs was persistently less accessible, indicating a chromatin failure to adequately respond to injury with aging (Fig. 1c).

In total, 1108 differentially accessible chromatin regions (DARs) (log2 FC ≤−1 or ≥1 and FDR ≤0.05) were identified in lung ECs between aged and young mice in absence of injury, where 82% (878) of them were less accessible and 18% (230) exhibited greater chromatin accessibility in aged lung ECs compared to young ones (Fig. 1d). We then evaluated DARs in injured lung ECs from young and aged mice relative to uninjured young lung ECs. In total, 3950 DARs were identified between injured vs uninjured young ECs (log2 FC ≤−1 or ≥1 and FDR ≤0.05), where 77% (3042) of them had increased chromatin accessibility and 23% (908) exhibited reduced chromatin accessibility in injured young ECs compared to uninjured ones (Fig. 1d). In addition, we found 3892 DARs between injured aged and uninjured young lung ECs, where 78% (3035) of them were less accessible and 22% (857) exhibited increased chromatin accessibility in injured aged lung ECs compared to uninjured young ones (Fig. 1d).

Remarkably, our ATAC-seq analysis also revealed that about 90% of DARs between young and aged lung ECs as well as following bleomycin-induced lung injury were detected within intragenic and intergenic genomic regions, and ~10% within promoters or untranslated regions (UTRs) (Fig. 1e). Altogether, these findings suggest that limited chromatin accessibility in lung ECs from aged mice may affect the expression of genes that control lung vascular repair following injury.

### Motif enrichment analysis identifies the transcription factor ERG as a putative regulator of lung vascular repair that is impaired in aging.
To identify putative lung endothelial transcription factors driving aging-associated chromatin alterations, we performed de novo motif analysis within DARs in aged vs young ECs using *Homer*[27]. Ranked by p value, the top predicted transcription factor was ERG (Fig. 2a), whose binding motif was present in more than 50% of DARs in lung ECs from aged mice compared to young (597 of 1108 total sites) (Fig. 2b). Among the

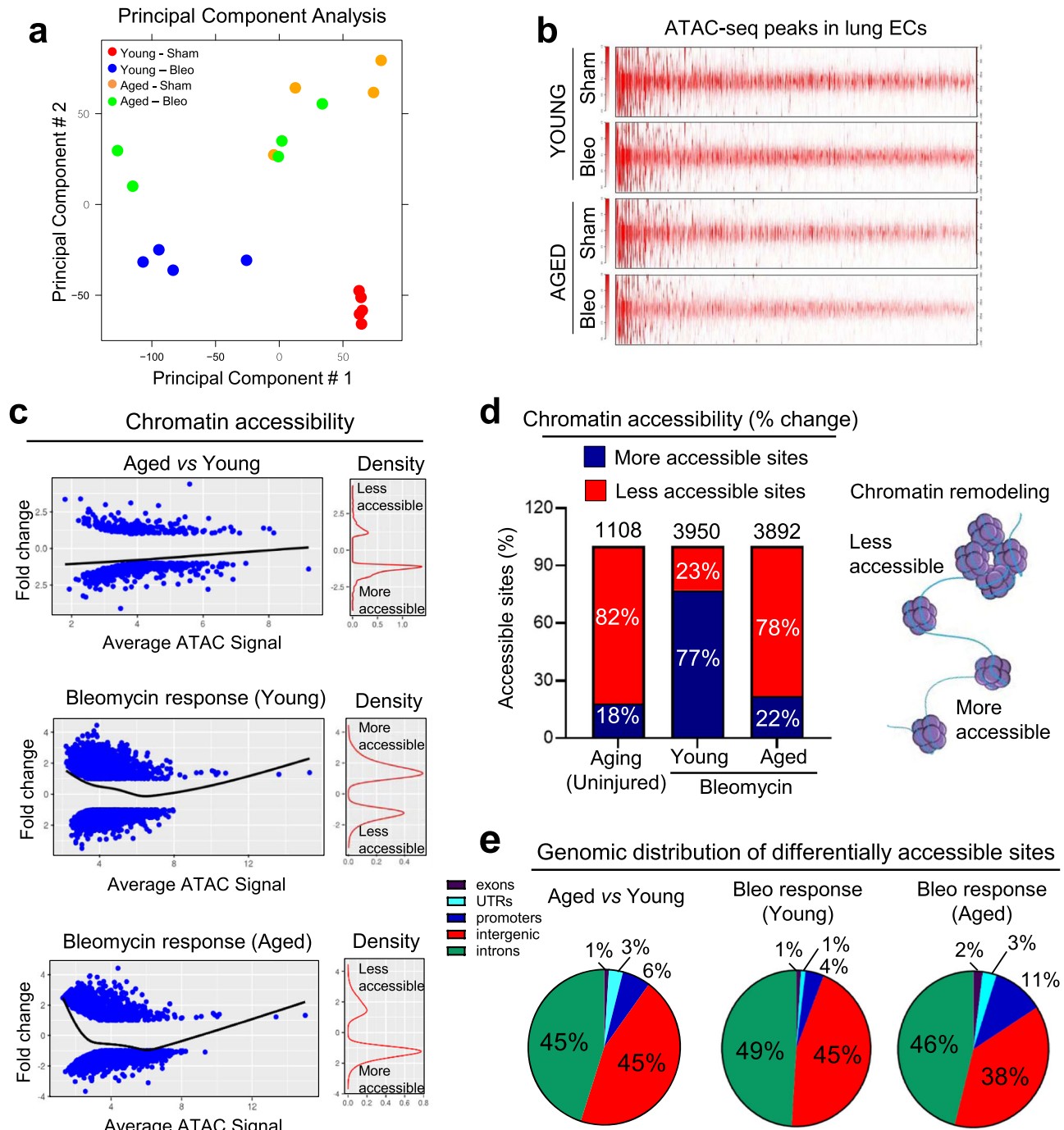

**Fig. 1 The landscape of chromatin accessibility in young and aged mouse lung ECs. a** Principal component analysis (PCA) of accessible loci as determined by ATAC-seq of freshly sorted lung ECs from young and aged mice, either Sham operated and 30 days after intratracheal Bleomycin delivery. Each dot represents an individual mouse (Young Sham 2-month-old, $N = 5$; Young Bleo 2-month-old, $N = 4$; Aged Sham 18-month-old, $N = 4$; Aged Bleo 18-month-old, $N = 5$). **b** Genome-wide ATAC-seq peaks in lung ECs. Each column represents one peak. The color represents the intensity of chromatin accessibility (dark red/more accessible and light red/less accessible). **c** Relative changes of chromatin accessibility in aged vs young ECs from sham mice, and in injured young and injured aged ECs vs young sham. Each dot represents one ATAC-seq peak. Black line in the left panel indicates average fold changes of peaks with the same ATAC-seq intensity. The density of open and closed peaks is shown under the density curve in the right panel. **d** Overall percentage of accessible chromatin sites in aged vs young ECs from sham mice, and in injured young and injured aged ECs vs young sham. **e** Genomic distribution of differentially accessible regions between young and aged lung ECs. The majority of differentially accessible sites were in introns and intergenic regions.

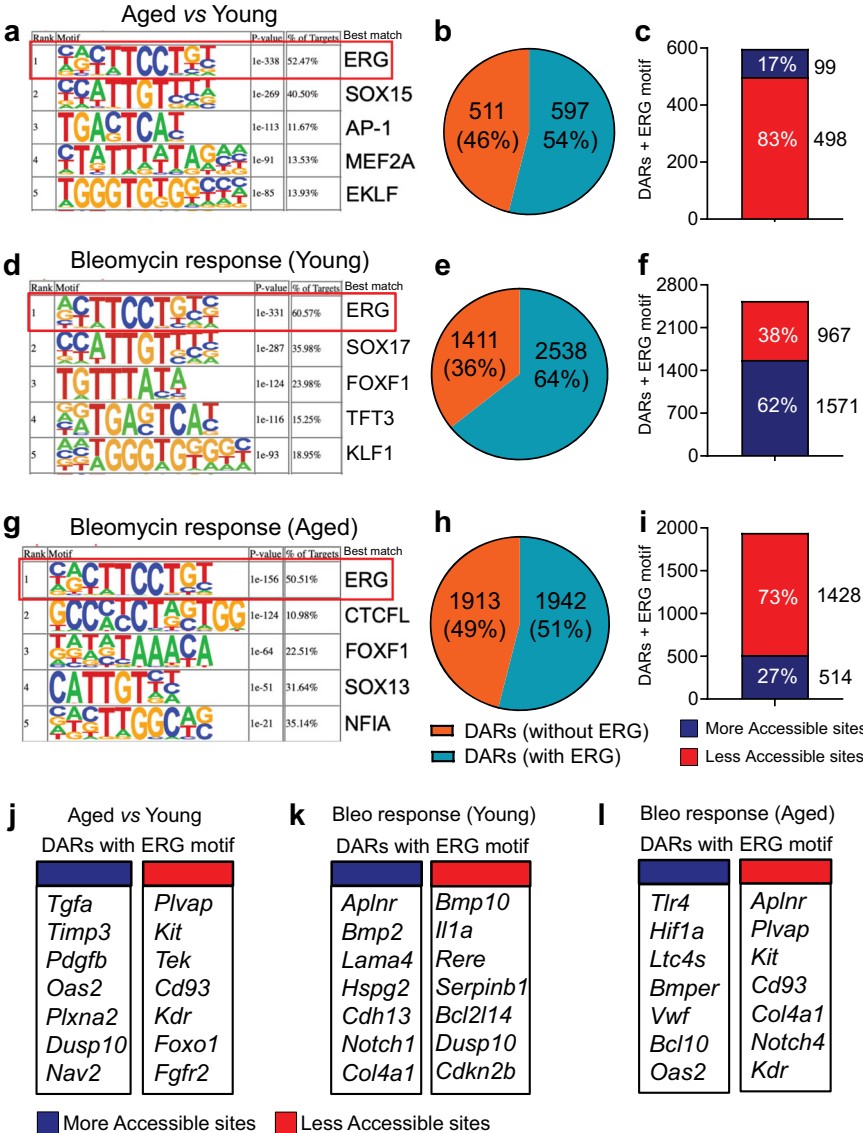

**Fig. 2 Motif analysis identifies ERG as a putative driver of lung vascular repair. a, b** De novo DNA motif analysis identifies ERG enrichment in more than 50% of differentially accessible chromatin sites in aged lung ECs relative to young ones. **c** In aged lung ECs ERG motif was associated with chromatin regions that exhibited reduced accessibility. **d, e** ERG enrichment was also observed in differentially accessible chromatin regions in young lung ECs following bleomycin challenge. **f** In young lung ECs ERG motif was associated with chromatin regions that exhibited increased accessibility. **g, h** Similarly to young lung ECs, ERG motif was also enriched in aged lung ECs following bleomycin challenge. **i** As opposed to young lung ECs, injured aged lung ECs exhibited ERG enrichment in chromatin areas with reduced accessibility. Motifs displayed in **a, d, g** were ranked by *p* value. **j** Representative ERG-target genes associated with differentially accessible chromatin regions in aged vs young lung ECs, and in injured young (**k**) and in injured aged lung ECs (**l**) relative to uninjured young lung ECs. ATAC-seq peaks associated with genes displayed in **j, k, l** were identified by a *p* value ≤0.05.

DARs containing ERG motifs, 83% (498 of 597 total sites) involved reduction of chromatin accessibility whereas only 17% (99 of 597 total sites) were associated with more accessible chromatin (Fig. 2c). Using the same approach, we discovered that ERG was also found to be highly enriched in more than 60% of DARs from injured young EC compared to uninjured ones (2538 of 3950) (Fig. 2d, e). Among them, 62% (1571 of 2538 total sites) were in chromatin regions with increased accessibility and 38% (967 of 2538 total sites) in regions with reduced chromatin accessibility (Fig. 2f).

Although the ERG motif was also largely present in DARs in injured aged lung ECs (Fig. 2g, h) (1942 of 3855), it was mainly associated with chromatin regions exhibiting reduced accessibility (73%, 1428 of 1942 total sites) (Fig. 2i). These findings suggest that aging affects the ability of ERG to engage lung endothelial

chromatin remodeling and exert its transcriptional function to effectively respond to injury.

Next, we used *Homer* to annotate the nearest transcriptional start site (TSS) to differentially accessible chromatin regions. This analysis revealed that lung ECs from aged mice exhibited reduced accessibility in numerous chromatin regions containing ERG motifs that were in close proximity to endothelial cell identity genes, including *Plvap, Kit, Tek*, and *Cd93*, as well as angiogenesis-related genes, such as *Foxo1, Kdr*, and *Fgfr2* (Fig. 2j). Of note, *Plvap, Kit, Tek*, and *Cd93* have been recently found to be distinctively expressed in gCap ECs[17], suggesting that ERG may regulate gCap EC homeostasis, and this intrinsic regulation is affected by aging.

Consistent with the role of ERG in the regulation of vascular function, we also found that transcriptionally active chromatin

regions in injured young lung ECs contained numerous ERG-target genes, including *Aplnr*, *Bmp2*, and *Notch1* (Fig. 2k). On the contrary, these angiogenesis-related genes were all associated with regions of reduced or unchanged chromatin accessibility in aged lung ECs, whereas fibrotic and inflammatory genes, such as *Tgfbr1*, *Oas2*, and *Ltc4s*, were associated with regions of increased chromatin accessibility (Fig. 2l). Our ATAC-seq analysis also revealed that multiple ERG-target genes encoding for basement membrane proteins, such as *Lama4* (Laminin 4), *Col4a1* (Collagen IV), and *Hspg2* (Perlecan) exhibited increased chromatin accessibility in young lung ECs but reduced or unchanged chromatin accessibility in aged lung ECs post-injury (Fig. 2k, l), suggesting that ERG-mediated transcription of basement membrane genes is diminished in aging. Thus, our ATAC-seq analysis demonstrates that normal vascular responses to lung injury are associated with increased chromatin accessibility along ERG-binding motifs in genes implicated in vascular remodeling, and these chromatin responses are impaired in aging. Thus, aging-induced changes in chromatin accessibility could affect the capacity of the lung microvasculature to repair the damaged lung parenchyma following injury.

**RNA-seq analysis identifies unique endothelial transcriptional signatures in the aged lungs.** To assess the impact that chromatin remodeling exerted on endothelial transcriptional responses, we carried out a parallel RNA-seq analysis on ECs isolated from the same lungs used for the ATAC-seq analysis. PCA analysis showed that ECs from injured and uninjured lungs shared the least similarities (Fig. 3a), indicating that injury strongly impacts the transcriptome of both young and aged lung ECs during tissue repair. We next determined the number of differentially expressed genes (DEGs) between young and aged lung ECs in the absence of injury. We identified 715 total genes (log2 FC $\leq -0.5$ or $\geq 0.5$, FDR $\leq 0.05$) that were differentially expressed between these two groups, with 465 genes upregulated and 250 genes downregulated in aged lung ECs compared to young ones. Ingenuity pathway analysis (IPA) revealed numerous canonical pathways that were affected in lung ECs from aged mice relative to young ones, and among them were those involved in angiogenesis (*Apelin signaling*), inflammation (*Complement System*), and oxidative stress (*NRF2 pathway*) (Fig. 3b).

Upstream regulator analysis identified putative mediators that may be responsible for the transcriptional signature observed in aged lung ECs. Among the top 10 upstream regulators were those involved in innate immune responses and inflammation, including IFN-γ, TNFα, STAT1, and TLR3, suggesting that aged lung ECs are characterized by enhanced innate immunity and inflammatory responses (Fig. 3c).

Consistent with these findings, we found numerous genes implicated in innate immunity responses, including *Cxcl10*, *Ifit1*, *Ltc4s*, *C3*, and *Serpina3n*, that were strongly increased in lung ECs from aged mice (Fig. 3d). Similarly, genes encoding for oxidative stress- and senescence-related proteins such as *Sod3*, *Nox4*, *Cdkn1a*, and *Pmaip1* were also upregulated as a result of aging (Fig. 3e). Consistent with our ATAC-seq data, we found that numerous angiogenesis-associated and ERG-target genes, including *Dll4*, *Mmp14*, *Sema3g*, and *Tek*, were strongly repressed in aged lung ECs relative to young ones (Fig. 3f), further suggesting that ERG-mediated angiogenic responses deteriorate with aging.

To assess whether changes in gene expression with aging were the direct result of an altered chromatin remodeling that may affect ERG-mediated transcription, we calculated the correlation of gene expression (RNA-seq cutoff: log2 FC $\leq -0.5$ or $\geq 0.5$ and $p$ value $\leq 0.05$) with chromatin accessibility (ATAC-seq cutoff: $p$ value $\leq 0.05$) followed by motif enrichment analysis (Fig. 3g).

Although we found that the overall changes in chromatin accessibility moderately correlated with gene expression [$R^2 = 0.254$ (Spearman)], the large majority of differentially accessible chromatin regions that contained ERG motifs (Q2 and Q3) positively correlated with gene expression (Fig. 3g). Strikingly, numerous angiogenesis-related genes as well as gCap EC marker genes followed this correlation, exhibiting both reduced chromatin accessibility and diminished gene expression with aging (Fig. 3h, i). On the contrary, the large majority of inflammatory and innate immunity genes whose expression increased with aging poorly or inversely correlated with chromatin accessibility (Supplementary Fig. 3). Interestingly, silencing of ERG in human lung microvascular endothelial cells (HLMECs) inhibited the expression of the vascular remodeling and cell identity genes *TEK*, *APLNR*, and *KIT* and upregulated the inflammatory genes *CXCL10*, *IFIT1*, *C3*, and *SERPINA3* (Fig. 3j, k), recapitulating the gene expression signatures observed in aged mouse lung ECs. Altogether, these data demonstrate that aging induces several transcriptional alterations and that aging-associated chromatin remodeling primarily impacted angiogenesis- and endothelial identity-related genes that are direct targets of ERG.

**RNA-seq analysis identifies maladaptive transcriptional responses of aged lung ECs to injury.** To elucidate transcriptional changes in lung ECs during the resolution or progression of bleomycin-induced lung fibrosis, we conducted differential gene expression analysis and identified genes that were uniquely upregulated, downregulated, or overlapping between young and aged lung ECs at day 30 after injury. As shown in Fig. 4a, we identified 387 (26.4%) and 468 (31.9%) genes that were uniquely upregulated in lung ECs from young and aged mice respectively, with 613 genes (41.8%) that were upregulated in both young and aged ECs post-bleomycin. We also identified 513 (45.6%) and 252 (22.4%) genes that were uniquely downregulated in young and aged lung ECs respectively, and 359 (31.9%) downregulated genes that overlapped between young and aged lung ECs post-injury.

Volcano plot analysis identified the top ten upregulated genes based on high significance and fold change in young lung ECs post-injury (Fig. 4b). Genes involved in the cell cycle (*Rgcc*), endothelial cell migration (*Fn1*, *Sparcl1*, *Mmp12*, and *Cxcl12*), and metabolism (*Pparg*) were significantly upregulated in injured young ECs compared to uninjured ones, suggesting that vascular remodeling takes place during the early phase of lung fibrosis resolution in young mice. Intriguingly, *Cxcl12* was recently identified as a gCap EC marker with a critical function in pulmonary capillary remodeling and alveolar regeneration[28], suggesting that gCap EC-mediated alveolar repair may play a role during lung fibrosis resolution.

Though numerous differentially expressed genes overlapped between lung ECs from young and aged mice post-injury, genes encoding for inflammatory and innate immunity mediators were among the top ten that were uniquely upregulated in aged lung ECs (Fig. 4c). Among them were *Serpina3n* and *Ltc4s*, both previously implicated in senescence and fibrosis[29,30]. IPA pathway analysis revealed that numerous angiogenesis pathways were enriched in young ECs post-injury, including "*Integrins*", "*Axonal guidance*", "*Apelin*", and "*Nitric oxide*" signaling pathways (Fig. 4d). On the contrary, inflammatory and senescence pathways, including, "*Leukocyte Extravasation*", "*Inflammasome*", and "*IL-6*" were highly enriched in aged lung ECs post-injury (Fig. 4e).

Given the contribution of vascular remodeling to lung fibrosis[16], we evaluated the expression of angiogenesis-related genes in lung ECs from injured young and aged mice relative to

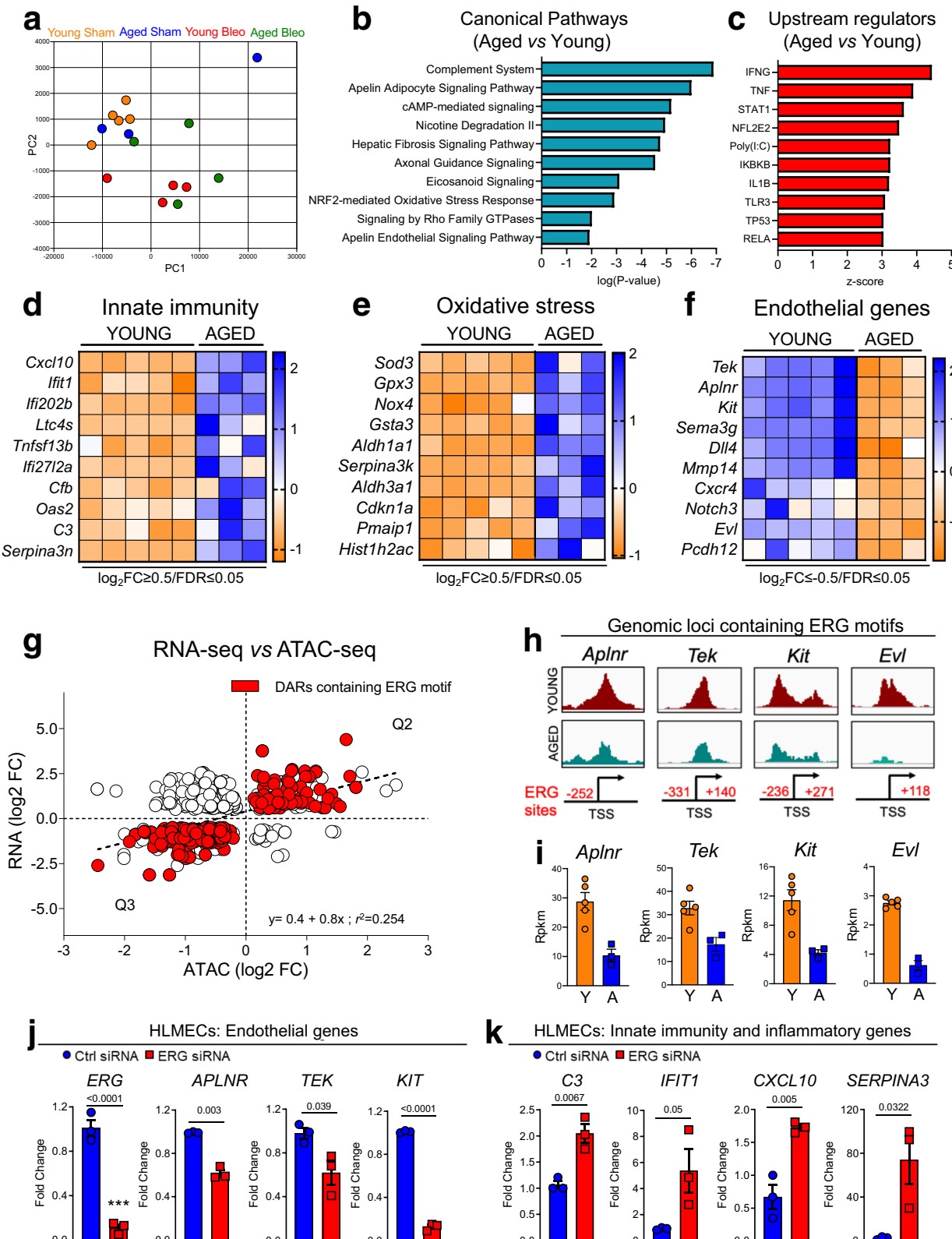

uninjured young ones. To this end, we focused on three distinct gene categories that feature in different phases of the angiogenesis process: "Proliferation", "Cytoskeleton & Cell migration", and "Basement membrane formation[31]". Although we did not observe significant differences in the expression of genes implicated in cell proliferation between injured young and injured aged lung ECs (Fig. 4f), the expression of genes implicated in cytoskeleton & cell migration, including *Cdh13*, *Evl*, *Nov*, and *Mmp14* (Fig. 4g), as well as basement membrane formation, such as *Lama4*, *Col4a1*, *Dcn*, and *Hspg2* (Fig. 4h), was strongly elevated in injured young ECs but not in injured aged ECs.

ATAC-seq and RNA-seq correlation together with motif enrichment analysis revealed that the large majority of differentially expressed genes positively correlated with chromatin

**Fig. 3 RNA-seq analysis identifies unique transcriptional signatures in lung ECs during aging. a** Principal components analysis (PCA) of RNA-seq data from freshly sorted lung ECs isolated from young and aged mice either sham or 30 following bleomycin (Young Sham 2-month-old, $N = 5$; Young Bleo 2-month-old, $N = 4$; Aged Sham 18-month-old, $N = 3$; Aged Bleo 18-month-old, $N = 4$). **b, c** Ingenuity pathway analysis shows canonical pathways and upstream regulators enriched in aged lung ECs relative to young ones. $P$ values were generated in IPA using Fisher's test ($\log_2$ FC $\leq -0.5$ or $\geq 0.5$, $p$ value $\leq 0.05$). $p$ value and activation $z$-score were used for plotting canonical pathways and activated upstream regulators respectively. **d–f** Heatmaps showing differentially expressed gene signatures (innate immunity, oxidative stress, and endothelial lineage) in aged lung ECs relative to young ones in absence of injury ($\log 2$ FC $\geq 0.5$ and FDR $<0.05$). A colored scale was used to display upregulated (blue) and downregulated (orange) genes. **g** Scatter plot showing the correlation between promoter accessibility and gene expression. Red dots indicate ERG targets. ATAC-seq peaks with a $p$ value $\leq 0.05$ and genes detected by RNA-seq with $\log_2$ FC $\leq -0.5$ or $\geq 0.5$ and FDR $\leq 0.05$ were plotted to compare the differentially accessible peaks and differentially expressed genes. Source data is provided as a Source Data file. **h** Genomic snapshots showing reduced chromatin accessibility of ERG-target genes with aging. **i** Histograms of RNA expression RPKM values showing reduced transcription of ERG-target genes with aging (Young Sham 2-month-old $n = 5$; Aged Sham 18-month-old $n = 3$). Values are summarized as mean and SD. Source data is provided as a Source Data file. **j, k** qPCR analyses of selected genes from RNA-seq demonstrated that ERG silencing in human lung microvascular ECs (HLMECs) recapitulated the gene expression signature observed in aging, including increased expression of inflammatory and innate immunity genes and reduced expression of endothelial genes ($N = 3$ independent experiments). Values are summarized as mean and SD and statistical analysis is performed using a two-tailed Student's $t$-test. Source data is provided as a Source Data file. $*P < 0.05$, $**P < 0.01$, $***P < 0.001$.

accessibility (Fig. 4i, j, Q2 and Q3) in injured young and aged lung EC contained ERG motifs in their promoters. Notably, although we found a positive correlation between gene expression and chromatin accessibility in both injured young [$R^2 = 0.334$ (Spearman)] and injured aged lung ECs [$R^2 = 0.294$ (Spearman)], the overall number of genes containing ERG motifs was reduced to approximately 50% in injured aged ECs compared to injured young ECs. Intriguingly, most of the genes implicated in cell proliferation poorly correlated with chromatin accessibility, as no differences were observed between injured young and injured aged lung ECs (Fig. 4k). In contrast, genes encoding for cytoskeleton/cell migration and basement membrane proteins were among the ERG-target genes that exhibited increased expression (Fig. 4g, h), and chromatin accessibility (Fig. 4l, m) in injured young ECs but were unchanged in injured aged ECs. Immunostaining of lung sections from young mice after injury showed that areas of active remodeling were characterized by increased vessel density and basement membrane formation as demonstrated by increased expression of CD31 and Collagen IV deposition (Supplementary Fig. 4A), whereas lungs from aged mice displayed vascular rarefaction and minimal Collagen IV protein deposition (Supplementary Fig. 4B).

These findings underline the maladaptive transcriptional response of the aged lung endothelium to injury and suggested that dysfunctional ERG signaling associated with abnormal chromatin accessibility in aging may limit endothelial migration, basement membrane formation, and ultimately vascular repair.

**ERG silencing in human lung ECs affects angiogenesis and enhances the secretion of fibrogenic mediators**. Angiogenesis is critical to support tissue repair following injury and its failure leads to unresolved injury and pathological matrix remodeling[32–34]. Previous studies demonstrated a critical role for ERG in angiogenesis[35–37]. To test the specific role of ERG in lung ECs during vascular assembly, we carried out a vasculogenesis assay in vitro using HLMECs[38]. As shown in Fig. 5a, ERG-silenced HLMECs failed to organize themselves into vascular networks compared to those treated with non-targeting siRNA, confirming the angiogenic function of ERG in lung ECs. In addition, ERG silencing impaired cell-cell contact and caused vascular disassembly as shown in Fig. 5b. Finally, the qPCR analysis demonstrated that ERG silencing in human lung ECs reduced the expression of genes involved in angiogenesis and vascular remodeling thereby recapitulating the gene expression signature of aged mouse lung ECs (Fig. 5c).

To test whether ERG-silenced lung ECs release inflammatory and fibrogenic mediators, we exposed normal human lung fibroblasts (nHLFs) to the conditioned medium (CM) derived from ERG-silenced or control ECs (Fig. 5d). CM from ERG-silenced cells strongly promoted fibroblast activation and enhanced the effect of the fibrogenic mediator TGFβ (Fig. 5e, f). Additionally, CM from ERG-silenced HLMECs strongly stimulated fibronectin and collagen I deposition by nHLFs (Fig. 5g). Consistent with these findings, analysis of CM using nanoscale liquid chromatography coupled to tandem mass spectrometry (nano LC-MS/MS) revealed increased secretion of numerous fibrogenic mediators by ERG-silenced ECs, including CTGF, Collagen Iα1, and PAI-1. In addition, in line with our in vivo experiments, basement membrane proteins such as Collagen IV and Laminin IV were reduced in the CM derived from ERG-silenced HLMECs (Fig. 5i). These findings demonstrate that ERG orchestrates lung angiogenic responses while inhibiting inflammatory and pro-fibrotic stimuli with paracrine effect on neighboring cells.

**Loss of endothelial ERG promotes pulmonary vascular remodeling and inflammation**. Preservation of vascular integrity is critical to limit tissue inflammation and to maintain organ homeostasis[39]. Excessive vascular remodeling, increased permeability, and inflammation have been shown to accompany numerous chronic diseases, including IPF[40]. To directly investigate the role of ERG in regulating pulmonary vasculature permeability in vivo, we generated conditional endothelial-specific *Erg* deficient mice (Cdh5-Cre- ER(T): Erg^flox/flox (ERG CKO), in which *Erg* gene is deleted from the endothelium upon tamoxifen administration (Fig. 6a). Daily doses of tamoxifen (total of 5 days) were delivered to ERG CKO and WT (Erg^flox/flox) mice followed by intravenous injections of Evans Blue to evaluate vascular leak 30 days after the last tamoxifen injection. As shown in Fig. 6b, vascular leak was markedly increased in the lungs of ERG CKO mice compared to WT mice, demonstrating that ERG is important in preserving lung vascular integrity. Since vascular leak is often accompanied by immune cell recruitment[41], we sought to investigate whether loss of endothelial ERG leads to increased immune cell infiltrations into the lungs. FACS analysis on whole lungs from ERG CKO and WT mice showed that loss of endothelial ERG exclusively enhanced lung neutrophil (CD45+, CD11b+, Ly6G+) infiltration (Fig. 6c and Supplementary Fig. 5), and this observation was confirmed in lung sections using an antibody against the neutrophil marker myeloperoxidase (MPO)[42] (Fig. 6d). Consistent with what we observed in aged mouse lung ECs, as well as in ERG-silenced human lung ECs, lungs from ERG CKO mice also exhibited elevated inflammation

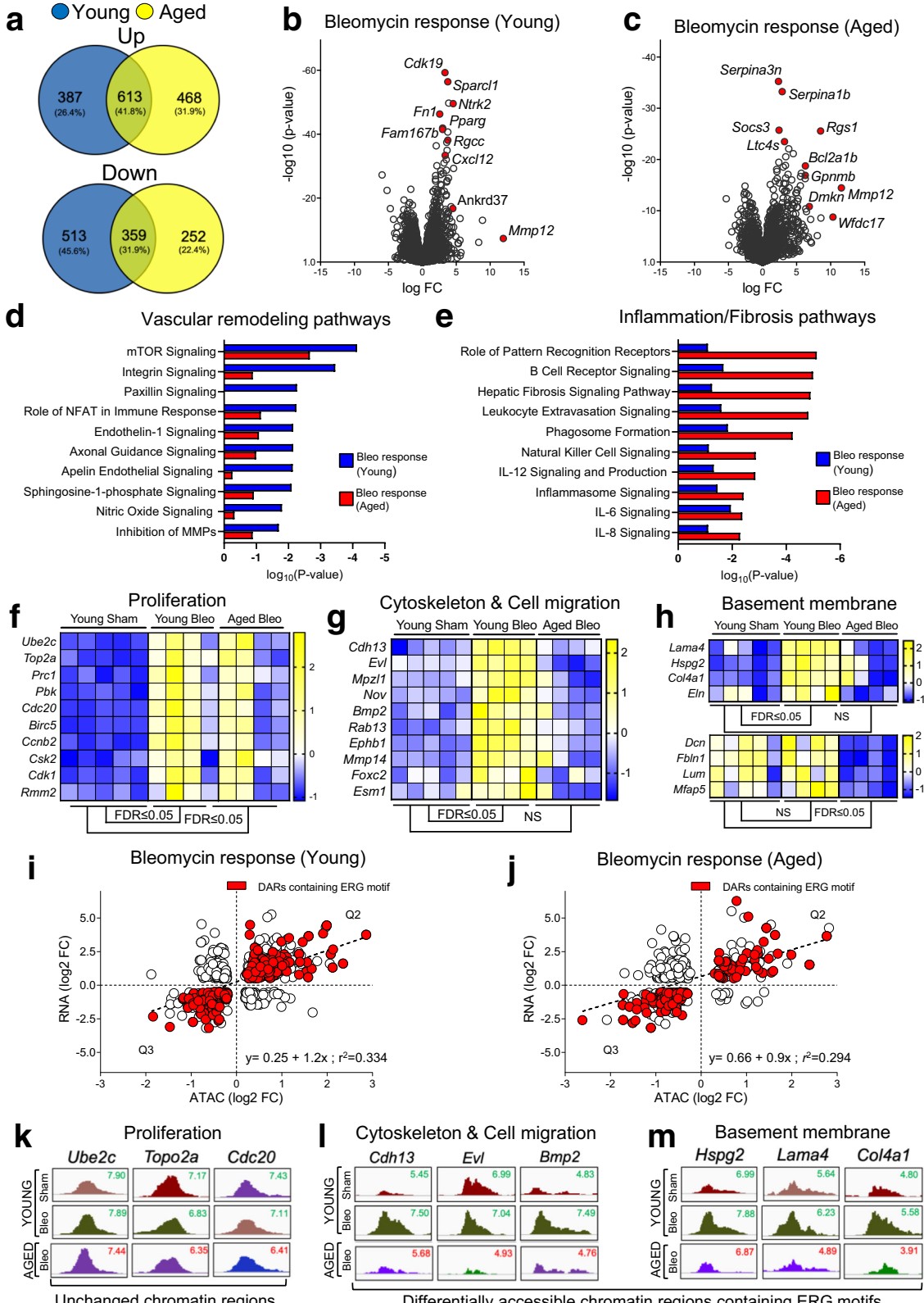

as demonstrated by the increased expression of inflammatory genes including *Il6*, *Cxcl10*, and *Tnfa* (Fig. 6e).

To assess whether ERG-deficient mice developed lung fibrosis, we measured hydroxyproline content followed by Masson's trichrome staining on WT and EGR CKO lungs at 45 days following the last dose of tamoxifen. Though we did not detect appreciable differences in hydroxyproline content between WT

and ERG CKO lungs (Fig. 6f), trichrome staining revealed that lungs from ERG CKO mice displayed aberrant perivascular remodeling mainly affecting bronchial arterial walls as well as alveolar capillaries (Fig. 6g), suggesting that loss of endothelial ERG may directly impact mural cells of the perivascular niche. These vascular abnormalities were also accompanied by red blood cell extravasation in the lung interstitium (data not shown), as

**Fig. 4 RNA-seq analysis identifies maladaptive transcriptional responses of aged lung ECs to injury. a** Venn Diagrams showing upregulated and downregulated genes in young and aged lung ECs following bleomycin delivery. **b, c** Volcano plots showing the distribution of differentially expressed genes based on fold changes and significance between young Bleo and young Sham and between aged Bleo and young Sham. Red dots represent the highest expressed and most significant genes in injured lung ECs compared to uninjured ones. **d, e** Pathway analysis identifies the top differentially regulated canonical pathways in injured young and injured aged lung ECs relative to uninjured ones. Vascular remodeling pathways were enriched in injured young lung ECs relative to uninjured ones whereas inflammation and fibrosis pathways were enriched in injured aged lung ECs relative to uninjured ones (data in the graphs are displayed as $p$ value). $p$ values for canonical pathways were generated in IPA using Fisher's test. Log($p$ values) $\leq -0.5$ or $\geq 0.5$ ($p$ value $\leq 0.05$) is being used for plotting noteworthy canonical pathways and activated upstream regulators respectively. **f–h** Heatmaps showing differentially expressed angiogenic gene signatures (proliferation, cytoskeleton & cell migration, and basement membrane) in young and aged lung ECs relative to uninjured young ones (log2 FC $\geq 0.5$ and FDR <0.05). A colored scale was used to display upregulated (yellow) and downregulated (blue) genes. **i, j** Scatter plots showing the correlation between promoter accessibility and gene expression. Red dots indicate ERG targets. RNA-seq cutoff: log2 FC $\leq -0.5$ or $\geq 0.5$ and FDR $\leq 0.05$, ATAC-seq cutoff: $p$ value $\leq 0.05$. Source data is provided as a Source Data file. **k** Genomic snapshots showing comparable chromatin accessibility of proliferation genes between young and aged lung ECs post-injury. **l, m** Genomic snapshots showing increased chromatin accessibility in ERG-target genes associated with vascular remodeling and angiogenesis in injured young EC. On the contrary, no changes in chromatin accessibility for these genes were observed in injured aged lung ECs.

previously reported[43]. This latter observation is consistent with the vascular leak and the altered vascular remodeling observed in ERG-deficient lungs.

**Loss of endothelial ERG impairs lung capillary regeneration and fibrosis resolution in young mice.** To evaluate the function of ERG on pulmonary vascular repair and lung fibrosis, we used bleomycin to induce lung fibrosis in both WT and ERG CKO mice. Bleomycin was delivered intratracheally followed by tamoxifen administration starting at day 14 post-injury to delete the *Erg* gene during the peak of fibrosis (Fig. 7a). Lungs were harvested at day 45 post-injury followed by weight determination and hydroxyproline content measurements to evaluate collagen deposition. As shown in Fig. 7b, c WT mice treated with both bleomycin and tamoxifen showed no difference in lung weight and hydroxyproline levels relative to tamoxifen-injected sham groups, confirming the self-resolving nature of this model. The lungs of bleomycin/tamoxifen-treated ERG CKO mice, however, still exhibited elevated weight and hydroxyproline content at the same time point, demonstrating that, similarly to aged mice, loss of endothelial ERG in young mice initiated at day 14 impaired lung fibrosis resolution following bleomycin challenge. These data were confirmed by H&E and Trichrome staining (Fig. 7d).

To further evaluate the fibrotic nature of ERG CKO lungs and confirm ERG deletion in the pulmonary vasculature, we immunostained injured WT and ERG CKO lungs using antibodies against ERG, CD31, and αSMA. We found that in WT lungs, blood vessels were enriched in areas with active tissue remodeling as demonstrated by their colocalization with cells expressing the myofibroblast marker αSMA (Fig. 7e). On the contrary, injured lungs from ERG CKO mice exhibited a reduced number of microvessels in areas of active fibrosis, suggesting that reduced ERG signaling in the lung capillary endothelium promotes vascular dysfunction and impaired fibrosis resolution. We also found that in WT but not in ERG CKO mouse lungs, areas of active remodeling exhibited abundant deposition of Collagen IV and Laminin IV, suggesting that ERG-mediated basement membrane protein secretion plays an active role during lung fibrosis resolution. Taken together, these findings demonstrated that ERG deletion in ECs of bleomycin-treated young lungs produces a phenotype that recapitulates the vascular dysfunction and the persistency of fibrosis of the aged lungs under the same experimental conditions.

**ERG-deficient mouse lungs manifest capillary abnormalities and fibrogenic features resembling aged and IPF lungs.** Studies using scRNA-seq have shown that both mouse and human lung ECs are transcriptionally heterogeneous[17–19,44], and identified two distinct populations of pulmonary capillary ECs, gCap ECs, and aCap ECs (aerocytes), with gCap ECs being important orchestrators of lung homeostasis and repair after injury[17]. Given the contribution of ERG to alveolar-capillary repair and fibrosis resolution in young mice, and the diminished expression of ERG-targeted genes with aging, we wondered whether loss of endothelial ERG would affect capillary EC heterogeneity. To address this question, scRNA-seq was performed on the lungs of WT and ERG CKO mice (Fig. 8a) following tamoxifen injection (90 days after the last tamoxifen injection). After quality filtering, we obtained 5655 cell profiles, 2783 from WT and 2872 from ERG CKO mice. Analysis of representative marker genes identified clusters of endothelial, epithelial, and mesenchymal cells, as well as hematopoietic cells including macrophages, monocytes, neutrophils, dendritic cells, natural killer cells, and B and T cells (Supplementary Fig. 6). To characterize the effect of *Erg* deletion on endothelial cell heterogeneity at a higher resolution, we re-clustered the ECs and removed all other cells from the analysis. Based on previously discovered gene markers[17], we identified ECs from the artery, vein, and lymphatic, as well as the newly identified capillary populations aCap ECs and gCap ECs (Fig. 8b). We also identified a new cluster of ECs in the lung of ERG CKO that was transcriptionally distinct from other clusters, which we called "intermediate ECs" (Fig. 8c, d). Interestingly, our single-cell transcriptomic analysis revealed a reduction of gCap ECs in the lungs of ERG CKO mice compared to WT mice (Fig. 8e). This reduction of gCap ECs was accompanied by an increased number of intermediate ECs (Fig. 8e), suggesting that ERG controls gCap EC identity, and that dysfunctional ERG signaling in these cells may affect their stemness. Our scRNA-seq analysis also revealed that non-vascular cells, including immune cells (macrophages and neutrophils), and mesenchymal cells (interstitial fibroblasts), were also affected by the loss of endothelial ERG. We found that the genes encoding for fibrogenic mediators previously implicated in IPF, such as *A100a9*, *Igf1*, *Ccl6*, and *Mmp9*, were elevated in lung immune cells from ERG CKO lungs compared to WT ones (Supplementary Fig. 7A, B). Intriguingly, the myofibroblast markers *Tnc*, *Ltbp2*, *Tagln*, and *Acta2* were elevated in lung fibroblasts from ERG CKO mice relatives to WT ones (Supplementary Fig. 7C). These data together with the abnormal perivascular manifestations of ERG CKO lungs, and the fibroblast activation induced by ERG-silenced human lung ECs, further reinforced the concept that dysfunctional endothelial ERG signaling in the aged lung and during fibrosis may enhance fibrogenic responses in bystander lung cells.

To further link dysfunctional ERG signaling to aging-associated vascular deterioration, we compared the transcriptome

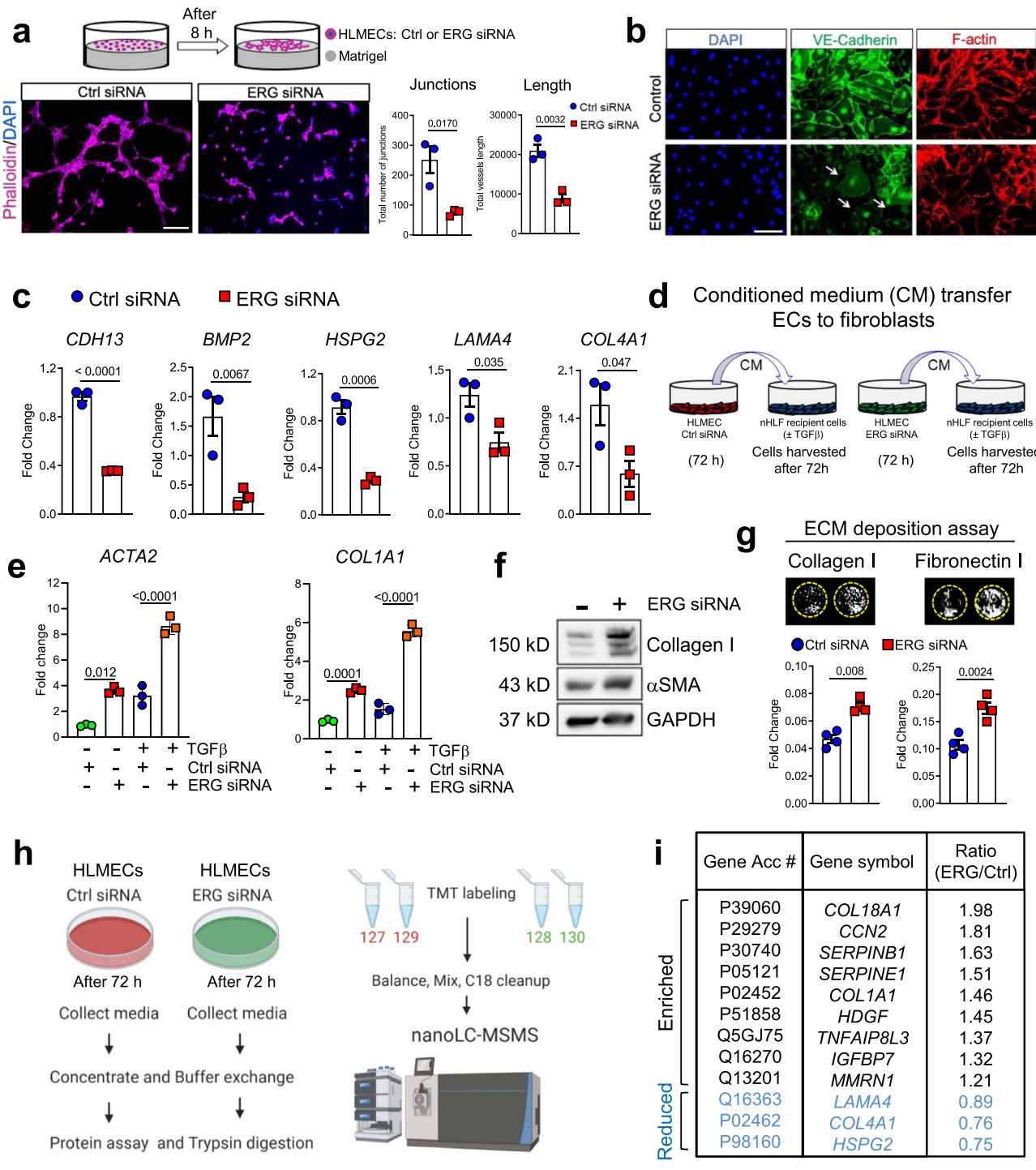

of the aged lung endothelium to that of ERG-deficient lung ECs. We found that 1671 genes were differentially expressed in aged lung ECs compared to young ones, whereas 2573 genes were differentially expressed in ERG-deficient ECs compare to WT ECs (Fig. 8g). Interestingly, we found 243 differentially expressed genes that were shared between the two comparisons, and among the upregulated ones, were numerous inflammatory and senescence genes, including *Cxcl10*, *Gpx3*, *Ifi27l2a*, and *Cdkn1a*. Notably, among the shared downregulated genes were the gCap EC markers *Aplnr*, *Kit*, *Tek*, *Sema3g,* and *Efna1*. These findings further support our premise that loss of ERG function recapitulates, at least in part, aging-associated transcriptional

abnormalities, including increased expression of inflammatory and senescence genes, and loss of gCap EC identity.

Given that ERG CKO mice, as well as aged mice, exhibited capillary abnormalities and impaired lung fibrosis resolution post-injury, we postulated that dysregulation of ERG in ECs may underpin IPF. Using a publicly available scRNA-seq dataset of normal and IPF lungs[23], we generated a de novo cluster analysis and discovered that, similarly to the lungs of ERG CKO, IPF lungs exhibited a drastic reduction of gCap ECs compared to healthy lungs (Fig. 8h–j). Of note, aCap ECs were also reduced, while pulmonary venous and systemic venous ECs were increased in IPF lungs (Fig. 8j), as recently reported[44]. Immunofluorescence

**Fig. 5 ERG silencing in HLMECs cells impairs angiogenesis and enhances the secretion of fibrogenic mediators. a** Effect of ERG-siRNA on VEGFA induced in vitro angiogenesis. Representative fields are shown. Quantitative analysis of tube formation was indicated as the total number of junctions and total vessel length. Image analysis in the whole photographed area were performed by using the software AngioTool. $N = 3$ independent experiments. Values are summarized as mean and SD and analyzed using a two-tailed Student's $t$-test. Source data is provided as a Source Data file. Scale bar: 50 µm. **b** Immunofluorescence staining shows reduction of VE-Cadherin and loss of cellular junctions (arrows) in ERG-silenced HLMECs compared to control cells. A representative image of $n = 3$ independent experiments is shown. Scale bar: 50 µm. **c** Quantitative PCR analysis in ERG-silenced HLMECs compared to control cells. $N = 3$ independent experiments. Values are summarized as mean and SD and analyzed using a two-tailed Student's $t$-test. Source data is provided as a Source Data file. **d** HLMECs were transfected prior to CM collection and transferred to recipient nHLFs. **e** qPCR analysis of recipient nHLF exposed to ERG-siRNA derived CM compared to cells exposed to control siRNA derived CM ($N = 3$ independent experiments). Values are summarized as mean and SD and analyzed using a one-way analysis of variance followed by Tukey's post hoc test. Source data is provided as a Source Data file. **f** Western blotting analysis of nHLs recipient cells exposed to ERG-siRNA derived CM compared to control siRNA derived CM. $N = 1$ experiment. **g** ECM deposition assay in nHLFs exposed to ERG-siRNA derived CM compared to cells exposed to control siRNA derived CM ($N = 4$ independent experiments, analyzed using two-tailed Student's $t$-test). Values are summarized as mean and SD. Source data is provided as a Source Data file. **h** Workflow employed for the analysis of the secretome in CM collected from control or ERG-silenced HLMECs (created with BioRender.com). **i** List of secreted proteins from control or ERG-silenced HLMECs ranked by their abundance ratio ERG/control. CM from three independent experiments was pulled and subjected to mass spectrometry. Source data is provided as a Source Data file. *$P < 0.05$, **$P < 0.01$, ***$P < 0.001$.

and transcriptional analysis confirmed the reduced number of gCap EC and the inhibition of several gCap EC markers in fibrotic human lungs compared to healthy ones (Fig. 8k, l).

Taken together, these findings suggest that limited ERG activity in fibrotic aged lungs may result in dysfunctional gCap EC turnover, loss of cell identity, and ultimately capillary dropout.

## Discussion

In this study, we provide new insights into lung endothelial epigenetic and transcriptional abnormalities driving vascular dysrepair and perpetuating pulmonary fibrosis in aging.

Chromatin remodeling is essential to coordinate gene transcription and its dysregulation often occurs with aging[45]. Little is known about how aged lung ECs respond to injury and how they integrate injury-mediated signals and chromatin remodeling into transcriptional programs that promote lung vascular repair following injury.

Here, we showed that aging was characterized by a widespread reduction of chromatin accessibility in lung ECs resulting in limited expression of genes involved in angiogenesis and vascular remodeling. Changes in chromatin accessibility in aged lung ECs also constrained bleomycin-induced vascular responses including the transcription of key genes implicated in vascular repair. On the contrary, ECs from young lungs exhibited an overall increase of chromatin accessibility and elevated transcription of vascular remodeling genes following lung injury. We identify the transcription factor ERG as a putative orchestrator of endothelial chromatin remodeling following lung injury, and whose dysfunctional signaling in aging contributed to aberrant endothelial transcription and lung dysrepair (Supplementary Fig. 8). Consistent with these findings, a recent paper reported that endothelial ERG binds the acetyltransferases p300, an epigenetic modifier and positive regulator of gene transcription, to drive the expression of vascular remodeling genes[46]. These findings together with our observations suggest that ERG regulates vascular chromatin remodeling following lung injury, potentially through epigenetic modifications and that these functions are impaired with aging.

Intriguingly, although inflammatory stimuli have been shown to inhibit the expression of endothelial ERG in vitro[21,47], and inflammation precedes fibrosis in the bleomycin model[48], our RNA-seq analysis did not show reduced expression of *Erg* gene in either injured or uninjured aged lung ECs compared to young ECs. These findings suggest that ERG post-translational modifications may be paramount for its efficient chromatin interaction and gene transcription. In this regard, a previous study demonstrated that Angiopoietin-1/TEK/AKT signaling pathway

in ECs stimulated ERG phosphorylation, and this modification is essential for its chromatin binding[49]. Interestingly, the expression of *Tek* gene was reduced in aged lung ECs compared to young lung ECs, and previous reports showed that the angiopoietin-1/TEK pathway deteriorates with aging[50,51]. In addition, reduced levels of Angiopoietin-1 were also reported in the serum of patients with IPF[52], suggesting that dysfunctional TEK signaling in fibrosis may contribute to aberrant ERG activity. On the other hand, since abnormal chromatin remodeling has been associated with aging[53], and our data demonstrated reduced endothelial chromatin accessibility with aging, we postulate that aging-associated chromatin remodeling may also affect ERG/DNA binding and transcription[54].

Our multi-omics analysis identified numerous ERG-target genes whose chromatin accessibility and expression were increased in young lung ECs but were either unchanged or reduced in aged lung ECs following injury. Among them, *Aplnr and Bmp2*, which encode Apelin receptor and Bone morphogenic protein-2 respectively, have been shown to promote angiogenesis and organ regeneration[55,56]. Interestingly, Apelin signaling exhibited an age-dependent decline in multiple organs and loss of its receptor leads to accelerated organ aging[57]. We also discovered that aged lung ECs and ERG-depleted lung ECs exhibited reduced expression of genes encoding for basement membrane proteins including Laminin IV and Collagen IV[58]. Besides playing an active role in angiogenesis[59], basement membrane proteins directly contribute to preserving vascular integrity and function in multiple organs including lungs[60]. In addition, Collagen IV participates in the formation of a mature basement membrane during alveologenesis[61] and influences alveolar epithelial cell regeneration[62], suggesting that reduced expression of basement membrane proteins in aging and during vascular repair may impair vascular and epithelial regeneration.

Tissue injury is typically accompanied by angiogenesis, which is key to reestablish normal homeostatic conditions[34]. Defective vascular regeneration, however, is often observed in elderly individuals and it is thought to impair tissue repair following injury[63]. Previous reports have shown that ERG regulates angiogenesis through the activation of multiple signaling pathways[36]. In line with this evidence, we demonstrated that ERG silencing in lung ECs inhibited angiogenesis and promoted the release of inflammatory and fibrogenic mediators in vitro. In addition, using our ERG CKO mouse model, we demonstrated that loss of endothelial ERG caused pulmonary perivascular remodeling, vascular leak, and inflammation. Strikingly, endothelial ERG depletion right after the inflammatory phase post-bleomycin challenge impaired lung vascular repair and fibrosis

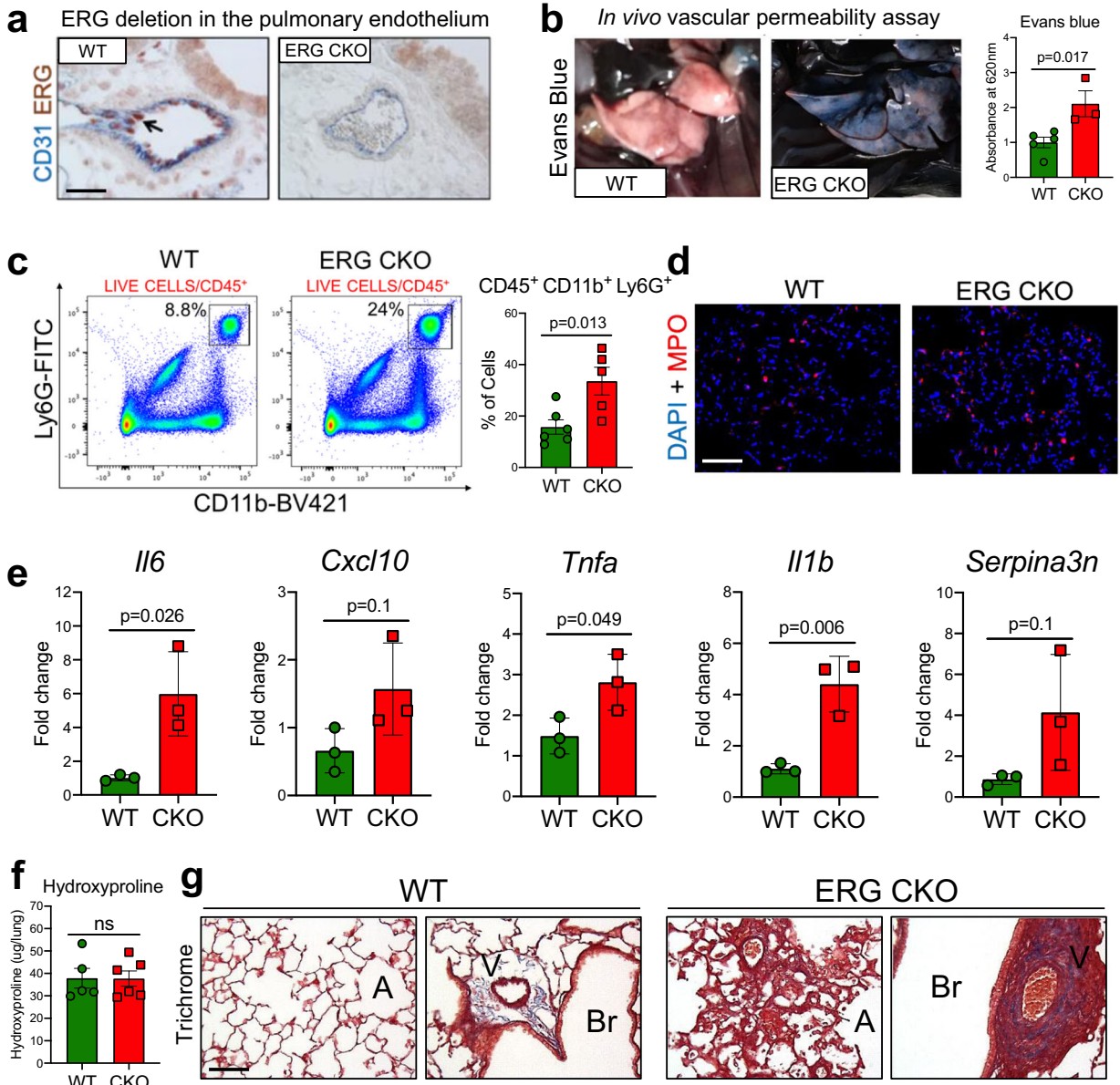

**Fig. 6 Conditional deletion of ERG in ECs promotes pulmonary vascular leak, vessel remodeling, and lung inflammation. a** Representative images of lung sections from control and Erg CKO animals (young mice, 2-month-old) after tamoxifen injections, stained with an anti-ERG antibody (red) and co-stained with an anti-CD31 antibody (blue) to evaluate Erg deletion in lung ECs (WT $n = 6$; CKO $n = 5$). Scale bar: 20 μM. **b** Vessel permeability assessed by i.v. injection of Evans blue (WT $n = 5$; CKO $n = 3$). Values are summarized as mean and SD and analyzed using a two-tailed Student's $t$-test. Source data are provided as a Source Data file. **c** FACS analysis of the whole CD45+ population and the overall fractions of CD45+/CD11b+/Ly6G+ neutrophils in mouse lungs from WT and ERG CKO mice (WT $n = 6$; CKO $n = 5$). Values are summarized as mean and SD and analyzed using a two-tailed Student's $t$-test. Source data are provided as a Source Data file. **d** Representative images of mouse lung sections were captured from WT and ERG CKO animals (WT $n = 6$; CKO $n = 5$). DAPI (blue) and the neutrophil marker MPO (red). Scale bar: 20 μm. **e** Transcriptional analysis of whole lung homogenates obtained from WT and ERG CKO mice (WT $n = 3$; CKO $n = 3$). Values are summarized as mean and SD and analyzed using a two-tailed Student's $t$-test. Source data are provided as a Source Data file. **f** Hydroxyproline assessment in WT and ERG CKO animals in the absence of injury (45 days after last tamoxifen injection) (WT $n = 5$; CKO $n = 6$). Values are summarized as mean and SD and analyzed using a two-tailed Student's $t$-test. Source data is provided as a Source Data file. **g** Representative images of Masson's Trichrome staining in lung sections from control and Erg CKO mice in the absence of injury, 45 days after last tamoxifen injection (WT $n = 5$; CKO $n = 6$). Scale bar: 100 μM.

resolution in young mice. Taken together these data suggest that compromised ERG-mediated chromatin remodeling and transcription in aging may affect angiogenesis and vascular recovery following lung injury leading to unresolved fibrosis.

Although in this paper we have mainly focused on the loss of angiogenic features and vascular repair properties by ECs with limited ERG functions, and during aging, our multi-omics analyses also revealed that numerous innate immunity pathways, including IFNγ, STAT1, TNFα, and TLR3, were enriched in aged mouse lung ECs. Notably, previous studies have shown that ERG inhibits the transcription of inflammatory and interferon genes in ECs of non-lung origin in vitro by limiting the transcriptional activity of STAT1 and NF-kB[47,64]. Consistent with these findings, aged lung ECs and ERG silencing in lung ECs upregulated

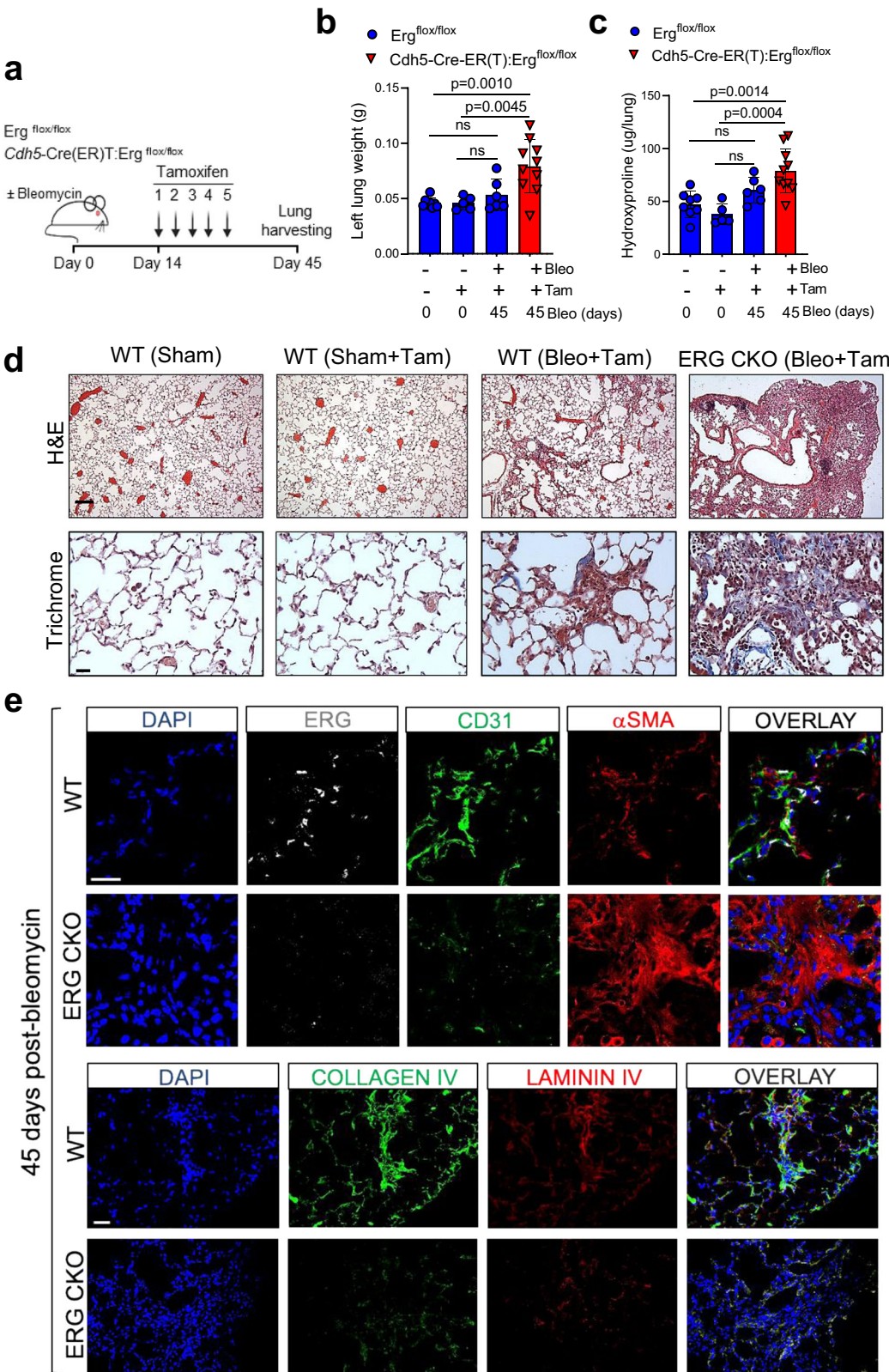

numerous genes downstream of STAT1 and NF-kB, including *CXCL10, IFIT1, C3,* and *SERPINA3*[65] and our proteomic analysis confirmed the secretion of numerous inflammatory and fibrogenic mediators by ERG-silenced lung ECs compared to control ECs. Consistently, our scRNA-seq data and lung tissue analysis showed increased lung fibroblast activation and immune cell recruitment in ERG CKO mice compared to WT ones. Intriguingly, previous reports have shown that loss of endothelial ERG led to endothelial-to-mesenchymal transition, spontaneous liver fibrogenesis[20], as well as aberrant activation of cardiac fibroblasts[22]. In addition, increased lung neutrophils and elevated inflammation have been often associated with cellular senescence,

**Fig. 7 Loss of endothelial ERG impairs lung capillary regeneration and fibrosis resolution in young mice. a** Schematic showing the experimental workflow to evaluate the effect of ERG deletion on bleomycin-induced lung fibrosis. **b, c** Lung weight and hydroxyproline assessments in WT and in ERG CKO animals (young mice, 2-month-old). Values are summarized as mean and SD and analyzed using one-way analysis of variance (followed by Tukey's post hoc test) (WT $n = 8$; WT + tamoxifen $n = 5$; WT + bleomycin+tamoxifen $n = 6$; CKO + bleomycin+tamoxifen $n = 10$). Source data are provided as a Source Data file. **d** Representative images of H&E (scale bar: 100 μM) and Masson's Trichrome (scale bar: 50 μM) staining in lung sections from WT and ERG CKO mice 45 days after bleomycin delivery (WT $n = 8$; WT + tamoxifen $n = 5$; WT + bleomycin+tamoxifen $n = 6$; CKO + bleomycin+tamoxifen $n = 10$). **e** Representative images of mouse lung sections were captured from control and Erg CKO animals. Top panel: DAPI (blue), ERG (white), CD31 (green), and αSMA (red); bottom panel: DAPI (blue), Collagen IV (green), Laminin IV (red) (WT $n = 8$; WT + tamoxifen $n = 5$; WT + bleomycin +tamoxifen $n = 6$; CKO + bleomycin+tamoxifen $n = 10$). Scale bar 20 μM.

aging, and IPF progression[66,67], suggesting that dysfunctional endothelial ERG signaling in IPF may contribute to elevated lung inflammation, increased senescence, and fibrosis progression.

Furthermore, our scRNA-seq analysis revealed, for the first time, that loss of endothelial ERG greatly impacted the transcriptome and the overall number of gCap ECs. While still preliminary, these findings implicated ERG as a putative regulator of gCap EC homeostasis whose function deteriorates with aging. Recently published scRNA-seq and tissue imaging data also revealed abnormal endothelial transcriptional signatures and an altered vascular circulation in human lungs from individuals with IPF[23]. Following the interrogation of publicly available scRNA-seq datasets from IPF and healthy lungs, we found that similar to what we observed in ERG CKO mice, IPF lungs exhibited a reduced number of gCap ECs along with reduced expression of gCap ECs markers compared to healthy lungs, suggesting that loss of beneficial endothelial transcriptional programs in the IPF vasculature may impair vascular regeneration and exacerbate fibrosis progression.

In conclusion, our results shed light on the role of pulmonary ECs in the progression of lung fibrosis, shifting their role from passive bystander to an active driver of fibrosis. Loss of chromatin homeostasis in the vasculature of the fibrotic aged mouse or IPF lungs may contribute to dysfunctional repair and progressive fibrosis. Identifying specific epigenetic alterations that can affect chromatin remodeling and lung vascular repair will be critical to develop strategies aimed at restoring aged pulmonary vasculature to a more youthful and healthy state that actively contributes to slow disease progression.

## Methods

**Mice and breeding**. Female and male Col1α1-GFP transgenic mice (2 and 18 months old, FVB strain) were generated as previously described (UC San Diego, La Jolla, CA)[68] and kindly provided by Dr. Derek Radisky. Female and male conditional Cdh5-CreER(T)-Erg[fl/fl] mice on a C57/BL6 background were generated by breeding Cdh5-CreER(T) mice (13073, Taconic, Rensselaer, NY, USA) with Erg[fl/fl] mice (STOCK Erg[tm1.1/wamo]/J, Jackson Laboratory, Bar Harbor, ME). Endothelial Erg deletion was induced by tamoxifen injection (5 injections of 0.5 mg, daily). All mice had access to food and water ad libitum and were on a 12 h/ 12 h light/dark cycle, ambient temperature 77–78 °F and humidity 46–49%.

**Cell culture**. Normal human lung microvascular endothelial cells (HLMECs) were purchased by Lonza (Walkersville, MD, USA, catalog # CC-2527) and maintained in Endothelial cell growth basal medium supplemented with microvascular endothelial cell growth medium SingleQuots (Lonza, Walkersville, MD, USA). In experiments involving siRNA, serum was reduced to 0.1%. All the experiments were performed with cells within the fourth passage. Normal primary human lung fibroblasts (nHLFs) were purchased by Lonza (Walkersville, MD, USA, catalog # CC-2512) and maintained in EMEM (ATCC, Manassas, VA, USA) containing 10% FBS. All the experiments were performed with cells between passages 3 and 7.

**Mouse model of bleomycin-induced lung injury**. All animal experiments were carried out under protocols approved by the Mayo Clinic Institutional Animal Care and Use Committee (IACUC) and by the Boston University IACUC and conforming to the ARRIVE guidelines. Bleomycin was delivered to the lungs as previously described[8,16]. Mice were anesthetized with ketamine/xylazine solution (100 and 10 mg/kg, respectively) and injected intraperitoneally. About 1 U/Kg bleomycin (APP Pharmaceutical, LCC Schaumburg, IL, USA) or PBS were

intratracheally delivered using a MicroSprayer (Penn-Century, Philadelphia, PA, USA). Body weight was monitored daily.

**FACS**. Mice were anesthetized with ketamine/xylazine solution (100 and 10 mg/kg, respectively), injected intraperitoneally, and perfused via the left ventricle with cold PBS 30 days after bleomycin or PBS delivery. The lungs were immediately harvested and minced with a razor blade in a 100 mm petri dish in a cold DMEM medium containing 0.2 mg/ml Liberase DL and 100 U/ml DNase I (Roche, Indianapolis, IN, USA). The mixture was transferred into 15 ml tubes and incubated at 37 °C for 35 min in a water bath under continuous rotation to allow enzymatic digestion. Digestion was inactivated with a DMEM medium containing 10% fetal bovine serum, the cell suspension was passed through a 40 μm cell strainer (Fisher, Waltham, MA, USA) to remove debris. Cells were then centrifuged (500×g, 10 min, 4 °C), and resuspended in 3 ml red blood cell lysis buffer (Biolegend, San Diego, CA, USA) for 90 s to remove the remaining red blood cells and diluted in 9 mL PBS after incubation. Cells were then centrifuged (500×g, 10 min, 4 °C) and resuspended in 0.2 ml of FACS buffer (1% BSA, 0.5 mM EDTA pH 7.4 in PBS).

For sorting, the single-cell suspension was then incubated with anti-CD45:PerCp-Cy5.5 (103132, clone 30-F11, Biolegend, San Diego, CA, USA, 1:200 dilution), anti-CD31:PE (102407, clone 390, Biolegend, San Diego, CA, USA, 1:200 dilution), anti-EpCAM:APC (118213, clone G8.8, Biolegend, San Diego, CA, USA, 1:200 dilution) antibodies, and DAPI (D3571, Thermo Fisher Scientific, Waltham, MA, USA, 1:1000 dilution) for 30 min on ice. After incubation, cells were washed with ice-cold FACS buffer and resuspended in 1 ml of FACS buffer. FACS sorting was conducted using a BD FACS Aria II (BD Biosciences, San Jose, CA, USA). To isolate CD45-, EpCAM-, GFP-, CD31+ population the following strategy was used: debris exclusion (FSC-A by SSC-A), doublet exclusion (SSC-W by SSC-H and FSC-W by FSC-H), dead cell exclusion (DAPI by PE), CD45 positive cell exclusion (PerCP-Cy5.5 by GFP), EpCAM and GFP positive cells exclusion (APC by GFP), and isolation of CD31 positive cells (APC by CD31). FACS-sorted ECs were subjected to mRNA and DNA isolation.

For analysis, the single-cell suspension derived from lungs of WT and ERG CKO mice, was obtained as described above and then incubated with anti-CD45:PerCp-Cy5.5 (103132, clone 30-F11, Biolegend, San Diego, CA, USA, 1:200 dilution), anti CD11b:BV421 (101235, clone M1/70, Biolegend, San Diego, CA, USA, 1:200 dilution) and anti Ly6G:FITC (127606, Biolegend, San Diego, CA, USA, 1:200 dilution), Ly6G:BV605 (128035, clone 1A8, Biolegend, San Diego, CA, USA, 1:200 dilution), NK1.1:PEcy5 (108715, clone PK136, Biolegend, San Diego, CA, USA, 1:200 dilution), SiglecF:APC-cy7 (565527, cloneE50-2440, BD Biosciences, San Jose, CA, USA, 1:100 dilution), F4/80:APC (123115, clone BM8, Biolegend, San Diego, CA, USA, 1:100 dilution), CD4:APC (100411, clone GK1.5, Biolegend, San Diego, CA, USA, 1:100 dilution), CD19:BUV395 (563557, clone 1D3, BD Biosciences, San Jose, CA, USA, 1:200 dilution), B220:PE (553090, clone RA3-6B2, BD Biosciences, San Jose, CA, USA, 1:200 dilution) and LIVE/DEAD cell stain kit (L23105, Thermo Fisher Scientific, Waltham, MA, USA, 1:1000 dilution). FACS analysis was performed with the following strategy: debris exclusion (FSC-A by SSC-A), doublet exclusion (SSC-W by SSC-H and FSC-W by FSC-H), and dead cell exclusion (DAPI by CD45). The immune cells were then characterized based on specific membrane markers as following: Neutrophils (CD45+, CD11b+, Ly6G +), NK cells (CD45+, NK1.1+, Ly6G+), Monocytes (CD45+, CD11b+, Ly6C+), Macrophages (CD45+, SiglecF+, F4/80+), Eosinophils (CD45+, CD11b+, SiglecF +), B cells (CD45+, CD19+, B220+), CD4+ T cells (CD45+, CD4+), CD8+ T cells (CD45+, CD8+). FACS analysis was conducted using a BD LSR II SORP (BD Biosciences, San Jose, CA, USA). Data were analyzed with FlowJo version 10.8.0 software (Tree Star Inc., Ashland, OR, USA).

**ATAC-seq and data analysis**. About 50,000 ECs were subjected to Omni ATAC-seq following the published protocol[69]. The size of library DNA was determined from the amplified and purified library by a Fragment Analyzer (Advanced Analytical Technologies; AATI; Ankeny, IA, USA), and the enrichment of accessible regions was determined by the fold difference between positive and negative genomic loci using real-time PCR. ATAC-seq data were analyzed using the HiChIP pipeline[70]. Paired-end sequencing reads were mapped to the mouse genome (mm10) using Burrows-Wheeler Aligner (BWA, v0.5.9). To remove read pairs mapping to

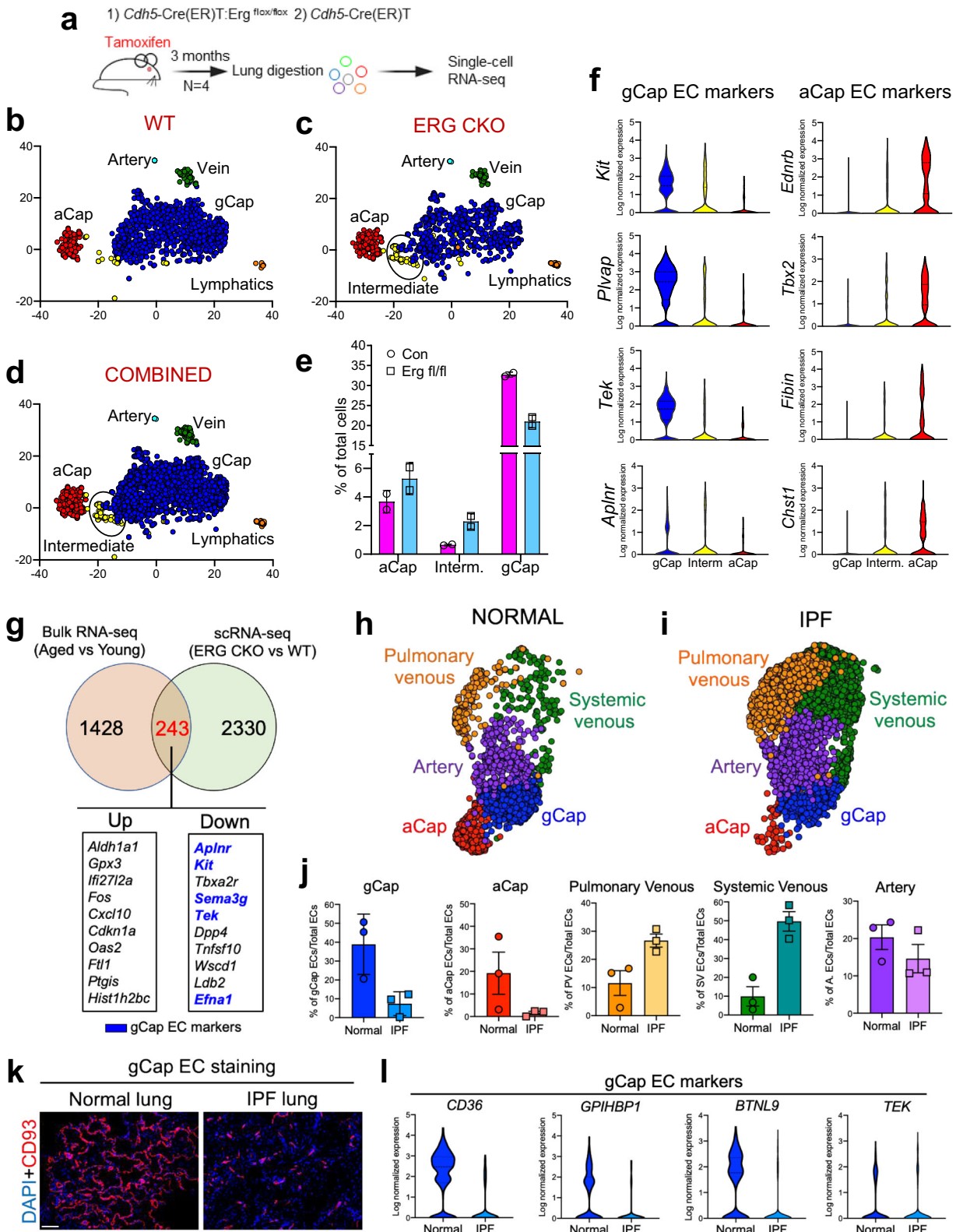

multiple locations in the genome, additional filtering using in-house scripts was performed, and only read pairs with one or both ends mapping uniquely to the genome were retained for downstream analysis. In addition, reads arising from PCR amplifications were discarded as duplicate reads using Picard MarkDuplicates (v1.67 http://broadinstitute.github.io/picard/). Open, accessible chromatin regions were identified using peaks detected by the software package "Model-Based analysis of ChIP-Seq" (MACS2, v2.0.10)[71]. ATAC-seq data standards and processing pipelines were conformed to and met the requirements of the ENCODE metrics (https://www.encodeproject.org/atac-seq/#standards). Statistically significant peaks

were selected for analysis using a false discovery rate (FDR) cutoff of ≤1%. Genes nearest to the peaks and the distance of peaks from transcription start sites (TSS) were annotated using Homer (http://homer.ucsd.edu/homer/index.html)[27]. For visualization of peaks across samples, normalized tag density profiles with a window size of 200 bp and a step size of 20 bp were generated using Bedtools (v2.16.2)[72] and in-house Perl scripts. Additionally, Tiled Data Format (TDF) tracks of the ATAC data were generated for visualization of the signals in Integrated Genomics Viewer (IGV, v2.8.13). For comparison and identification of differential peaks across conditions, the software package DiffBind (v2.14.0) was used at

**Fig. 8 ERG-deficient mouse lungs manifest capillary abnormalities and fibrogenic features resembling aged and IPF lungs. a** Schematic showing the experimental strategy to perform scRNA-seq analysis. **b–d** Wild type, Erg CKO, and combined t-SNE visualization to identify subpopulations. Colors denote corresponding clusters ($N = 2$). **e** Percent of aCAP, intermediate, and gCAP endothelial cells of total endothelial cells. Source data are provided as a Source Data file. **f** Violin plots of gCAP and aCAP gene expression in each endothelial cell population. **g** Venn Diagrams showing the comparison between the RNA-seq (Young vs Aged) and the scRNA-seq of ERG-deficient lung ECs (RNA-seq cutoff: log $_2$FC ≤−0.5 or ≥0.5 and FDR ≤0.05; scRNA-seq cutoff: log $_2$FC ≤−1 or ≥1 and FDR ≤0.001). **h, i** UMAP of endothelial cells from the publicly available dataset of normal and IPF lungs. **j** Percent of endothelial cell subpopulations of total ECs in normal ($n = 3$ independent samples) and IPF ($n = 3$ independent samples) lungs. Source data is provided as a Source Data file. **k** Representative images of human lung sections were captured from control and IPF lungs. DAPI (blue) and the human gCap EC marker CD93 (red). Scale bar 100 µM. **l** Violin plot of the expression of the human gCap EC markers *CD36, GPIHBPI, BTNL9,* and *TEK* in normal and IPF lungs. Values are summarized as mean and SD.

statistical significance thresholds of *p* value <0.05 and an absolute fold change value of greater than 1. Additional plots for visualization of differential peaks and chromosomal distribution of peaks were generated using in-house R scripts and the RCircos software package. Motif analysis was completed using the findMotifs-Genome.pl command within Homer (http://homer.ucsd.edu/homer/index.html)[27].

**RNA-seq and data analysis.** RNA quality was determined with a Fragment Analyzer instrument (Agilent, Santa Clara, CA). RNA samples that had RQN values ≥6 were approved for library preparation and sequencing. RNA libraries were prepared using 200 ng of total RNA according to the manufacturer's instructions for the TruSeq RNA Sample Prep Kit v2 (Illumina, San Diego, CA), employing poly(A) mRNA enrichment using oligo(dT) magnetic beads. The RNA integrity numbers (RIN) for each sample were: young sham: 8.3–9.7; young bleo: 9.5–10; aged sham: 7.9–9.5; aged bleo: 9.3–10. The final adapter-modified cDNA fragments were enriched by 12 cycles of PCR using Illumina TruSeq PCR primers. The concentration and size distribution of the completed libraries were determined using a Fragment Analyzer (Agilent) and Qubit fluorometry (Invitrogen, Carlsbad, CA). Libraries were sequenced following Illumina's standard protocol using the Illumina cBot and HiSeq 3000/4000 PE Cluster Kit, which 33-40 million fragment reads per sample. The flow cells were sequenced as 100 × 2 paired-end reads on an Illumina HiSeq 4000 using HiSeq 3000/4000 sequencing kit and HCS v3.3.52 collection software. Base-calling was performed using Illumina's RTA version 2.7.3. Data were analyzed using MAPRSeq version 3.0.2.[73], which is an integrated RNA-seq bioinformatics pipeline developed at the Mayo Clinic for comprehensive analysis of raw RNA sequencing paired-end reads. MAPRSeq employs STAR[74], a splice-aware, accurate and fast aligner for aligning reads to the reference human genome (build mm10). Aligned reads are then processed through multiple modules in a parallel fashion. Gene and exon expression quantification is performed using the Subread[75] package to obtain RPKM (Reads Per Kilobase of transcript, per Million mapped reads). Finally, comprehensive quality control modules from the RSeQC[76] package are run on aligned reads to assess the quality of the sequenced libraries. The bioinformatics package DESeq2 is used for differential gene expression analysis[77]. The criteria for selection of significant differentially expressed genes were: log2 fold change ≤−0.5 or ≥0.5 and Benjamini–Hochberg adjusted *p* value ≤0.05. This list of differentially expressed genes was used for investigating enriched canonical pathways and upstream regulators using Core analysis from the Ingenuity Pathway analysis (IPA, Ingenuity® Systems, www.ingenuity.com). *P* values were generated in IPA using Fisher's test (Log₂ fold change ≤−0.5 or ≥0.5, *p* value ≤0.05). *p* value and activation *z*-score were used for plotting noteworthy canonical pathways and activated upstream regulators respectively.

**Integration analysis of ATAC-seq and RNA-seq datasets.** The ATAC-seq peak data were compared to the RNA-seq data to determine how chromatin accessibility influenced gene expression. The differential log2 fold change by TMM normalized values for RNA-seq and ATAC-seq counts for each group comparison were associated using the gene annotation of the ATAC-seq TSS/Promoter peak with the assigned RNA-seq gene. ATAC-seq peaks with a *p* value ≤0.05 and genes detected by RNA-seq with log2 fold change ≤−0.5 or ≥0.5 and FDR ≤0.05 were plotted against each other to compare the differentially accessible peaks and differentially expressed genes.

**Single-cell RNA sequencing and data analysis.** Following tamoxifen injections, the lungs of two WT and two ERG CKO mice were harvested and disassociated into the single-cell suspension as described for FACS. Cell suspensions were brought to the Single-Cell Sequencing Core at the Boston University Medical Center for processing. Cell viability and counts were determined using Countess II Automated Cell Counter. Single cells, reagents, and a single Gel Bead containing barcoded oligonucleotides are encapsulated into nanoliter-sized Gel Bead-in-Emulsion using the 10x Genomics GemCode platform (10X Genomics, Pleasanton, CA, USA). Lysis and barcoded reverse transcription of RNAs from single cells is performed as described by 10x genomics. Enzyme fragmentation, A tailing, adapter ligation, and PCR are performed to obtain final libraries containing P5 and P7 primers used in Illumina bridge amplification. Size distribution and molarity of

resulting cDNA libraries were assessed via Bioanalyzer High Sensitivity DNA Assay (Agilent Technologies, USA). All cDNA libraries were sequenced on an Illumina NextSeq 500 instrument according to Illumina and 10X Genomics guidelines with 1.4–1.8 pM input and 1% PhiX control library spike-in (Illumina, USA). Sequencing data were processed using 10X Genomics' Cell Ranger pipeline to generate feature/barcode matrices from raw count data.

Feature and barcode matrices of WT and ERG CKO were imported into BioTuring Browser 2 (Bioturing, San Diego, USA) for analysis. After quality filtering, we obtained 5655 cell profiles, 2783 from WT and 2872 from ERG CKO mice. We perform t-Distributed Stochastic Neighbor Embedding (t-SNE) dimensionality reduction with canonical correlation analysis (CCA) subspace alignment and performed unsupervised graph-based clustering. Analysis of representative marker genes identified clusters of endothelial, epithelial, and mesenchymal cells as well as hematopoietic cells. Endothelial cells were selected and further re-clustered for analysis.

Publicly availably human IPF single-cell RNA-seq data[23] were downloaded from the BioTuring repository and endothelial cells were analyzed using the BioTuring Browser 2.

**Real-time PCR.** Total mRNA was isolated using RNeasy micro kit or minikit (Qiagen, Valencia, CA, USA) followed by Nanodrop concentration and purity analysis. cDNA was synthesized using SuperScript VILO (Thermo Fisher Scientific, Waltham, MA, USA); RT-PCR was performed using FastStart Essential DNA Green Master (Roche Diagnostics, Mannheim, Germany) and analyzed using a LightCycler 96 (Roche Diagnostics, Mannheim, Germany). RT-PCR primers used in this study (Integrated DNA Technologies, Coralville, IA, USA) are listed in Table 1.

**Fibrosis evaluation.** Hydroxyproline content was measured using a hydroxyproline assay kit (Biovision, Milpitas, CA, USA). Briefly, lung samples were transferred into glasses tubes and hydrolyzed with 200 µl 6 N HCL at 110 °C for 48 h. The hydrolyzed samples were evaporated to remove excess HCL, reconstituted with 400 ml H₂O and filtered in 1.5 ml centrifuge tubes equipped with a 0.45 µm semipermeable membrane filter. After samples were added to a 96-well microplate, Chloramine T solution was added to each well and the plate was incubated at 65 °C for 18 min. This method gives an orange-red color which is linear up to 6 µg of hydroxyproline. OD 550 nm was obtained and compared to a hydroxyproline standard curve.

**In vivo vascular permeability assay.** The lung microvascular permeability was measured in WT and ERG CKO animals using the Evans blue dye extravasation technique. Evans blue (20 mg/Kg, Sigma-Aldrich, St. Louis, MA, USA) was injected i.v. through the tail vein 30 min before sacrifice. Lungs were then harvested and the absorbance of Evans blue was measured at 620 nm.

**Immunohistochemistry.** Formalin-fixed paraffin-embedded (FFPE) tissues were cut in serial sections (7 µm). The FFPE sections were deparaffinized using a standard protocol of xylene and alcohol gradients. Sections were then blocked first with BLOXALL endogenous peroxide blocker (SP-6000-100, Vector Laboratories, Peterborough, UK) and then with 5% goat serum and 2% BSA (Sigma-Aldrich, St. Louis, MA, USA). The staining was performed by using the VECTASTAIN Elite ABC HRP kit (PK-6200, Vector Laboratories, Peterborough, UK), anti-CD31 rat primary antibody (550274, clone MEC 13.3, BD Biosciences, San Jose, CA, USA, 1:200 dilution), anti-ERG rabbit primary antibody (97249, Cell Signaling Technology, Danvers, MA, USA, 1:100 dilution) and detection with impact DAB (Vector Laboratories, Peterborough, UK). Slides were then dehydrated using a standard protocol and coverslipped using DPX mountant (Sigma-Aldrich, St. Louis, MA, USA). Masson's trichrome staining was performed by using a commercially available stain kit (HT15, Sigma-Aldrich, St. Louis, MA, USA). Hematoxylin and eosin (H&E) staining was performed by using hematoxylin solution (Sigma-Aldrich, St. Louis, MA, USA) and eosin solution (Millipore, Burlington, MA, USA).

**Table 1 Human and mouse primer sequences for qPCR analysis.**

| Primers | Forward | Reverse |
|---|---|---|
| GAPDH | GGAAGGGCTCATGACCACAG | ACAGTCTTCTGGGTGGCAGTG |
| APLNR | CTCTGGACCGTGTTTCGGAG | GGTACGTGTAGGTAGCCCACA |
| C3 | GGGGAGTCCCATGTACTCTATC | GGAAGTCGTGGACAGTAACAG |
| SERPINA3 | GCTCATCAACGACTACGTGAA | CACCATTACCCACTTTTTCTTGC |
| TEK | TTAGCCAGCTTAGTTCTCTGTGG | AGCATCAGATACAAGAGGTAGGG |
| IFIT1 | TTGATGACGATGAAATGCCTGA | CAGGTCACCAGACTCCTCAC |
| CFB | GCACTGGAGTACGTGTGTCC | CCCGTTCTCGAAGTCGTGTG |
| KIT | ACTTGAGGTTTATTCCTGACCCC | GCAGACAGAGCCGATGGTAG |
| CXCL10 | GTGGCATTCAAGGAGTACCTC | TGATGGCCTTCGATTCTGGATT |
| CADH13 | AGTGTTCCATATCAATCAGCCAG | CCTTACAGTCACTGAAGGTCAAG |
| EVL | CTTCCGTGATGGTCTACGATG | TGCAACTTGACTCCAACGACT |
| IL18 | TCTTCATTGACCAAGGAAATCGG | TCCGGGGTGCATTATCTCTAC |
| BMP2 | ACCCGCTGTCTTCTAGCGT | TTTCAGGCCGAACATGCTGAG |
| HSPG2 | CCAAATGCGCTGGACACATTC | CGGACACCTCTCGGAACTCT |
| LAMA4 | AGGATACTGTGTGACTACTGACG | TGAACGATAGGGTAGAAGCTGAA |
| COL4A1 | CCAGGGGTCGGAGAGAAAG | GGTCCTGTGCCTATAACAATTCC |
| Il6 | TAGTCCTTCCTACCCCAATTTCC | TTGGTCCTTAGCCACTCCTTC |
| Cxcl10 | CCAAGTGCTGCCGTCATTTTC | GGCTCGCAGGGATGATTTCAA |
| Tnfa | CCCTCACACTCAGATCATCTTCT | GCTACGACGTGGGCTACAG |
| Il1b | GCA ACT GTT CCT GAA CTC AAC T | ATC TTT TGG GGT CCG TCA ACT |
| Serpina3n | ATTTGTCCCAATGTCTGCGAA | TGGCTATCTTGGCTATAAAGGGG |

**RNA interference**. Transient RNA interference was performed with siGENOME non-targeting Control siRNA Pool #1 (D-001206-13-05, 25 nM) or siGENOME Human ERG-siRNA (L-003886-00-0005, 25 nM) by using Lipofectamine RNAi-MAX reagent (13778075, Thermo Fisher Scientific, Waltham, MA, USA). On the day of transfection, the medium was switched to 0.1 % FBS and the cells were harvested after 72 h.

**Immunofluorescence staining**. Cells or slides (tissue sections: 7 μM) were fixed in 3.7% formalin (Sigma-Aldrich, St. Louis, MA, USA), permeabilized in 1% Triton X-100 (Sigma-Aldrich, St. Louis, MA, USA), blocked with 5% BSA for 1 h and incubated with anti-CD31 rat primary antibody (550274, clone MEC 13.3, BD Biosciences, San Jose, CA, USA, 1:500 dilution), anti-Collagen IV rabbit primary antibody (NB120-6586, Novus Biologicals, Centennial, CO, USA, 1:200 dilution), anti-Laminin 4 goat primary antibody (NBP2-42392, clone CL3183, Novus Bio-logicals, Centennial, CO, USA, 1:200 dilution), anti-ERG rabbit primary antibody (97249, Cell Signaling Technology, Danvers, MA, USA, 1:500 dilution), anti αSMA mouse primary antibody (F3777, clone 1A4, Sigma-Aldrich, St. Louis, MA, USA, 1:200 dilution) or anti VE-Cadherin rabbit primary antibody (D87F2, Cell Sig-naling Technology, Danvers, MA, USA, 1:400 dilution), or anti MPO goat primary antibody (AF3667, R&D System, Minneapolis, MN, USA, 1:200 dilution), followed by fluorescence-conjugated secondary antibodies (Alexa Fluor 488-, Alexa Fluor 555- or Alexa Fluor 647- conjugated, Thermo Fisher Scientific, Waltham, MA, USA, 1:800 dilution) and DAPI (62248, Thermo Fisher Scientific, Waltham, MA, USA, 1:1000 dilution) to counterstain nuclei. Controls were done by omitting the primary antibodies.

**Vasculogenesis assay**. Control or ERG-silenced HLMECs were seeded in 96-well plates covered with polymerized growth factor reduced Matrigel matrix (10⁴ cells/well). After 8 h, capillary-like structures were fixed and stained with phalloidin and DAPI, to highlight the cytoskeleton and the nuclei, respectively. Cytoskeleton and nuclei were stained with phalloidin and DAPI, respectively. Quantitative analysis of tube formation, performed by using the software AngioTool[78], were indicated as total number of junctions and total vessel length.

**Mass spectrometry and data analysis**. Ten milliliters of conditioned media were concentrated to 200 ml using an Amicon Ultrafree 3 K cutoff centrifugal device (MilliporeSigma, Burlington, MA), buffer exchanged into 20 mM HEPES (pH 8), dried in a speed vac, and solubilized in 30 ml of 8 M urea in 20 mM HEPES (pH 8). Reduction and alkylation were performed with 5 mM Tris(2-carboxyethyl) phos-phine hydrochloride and 10 mM iodoacetamide, respectively. The samples were diluted to 2 M urea followed by digestion with sequencing grade modified trypsin (Promega, Madison, WI, USA) at a 1:20 ratio of enzyme to protein for 16 h at 37 °C. After digestion, desalting was carried out using TopTip $C_{18}$ spin tips (Glygen Corp, Columbia, MD, USA) according to the manufacturer's instructions. Samples were dried down, reconstituted in 100 mM triethylammonium bicarbonate buffer (pH 8.5), split into duplicates, and labeled with 100 mg of the TMT 6-plex isobaric reagents (Thermo Fisher Scientific, Waltham, MA) following the manufacturer's instructions with 1 h reaction at room temperature and quenching with 5%

hydroxylamine. Reagents with reporter ions corresponding to 127 and 129 were used for labeling ERG-siRNA samples while those corresponding to 128 and 130 were used for labeling control siRNA samples. The excess TMT reagents were removed from the pooled mixture using $C_{18}$ spin tips (Glygen Corp, Columbia, MD, USA) and the samples were analyzed on an Orbitrap Eclipse Tribrid mass spectrometer (Thermo Fisher Scientific, Waltham, MA, USA) coupled to an Ultimate 3000 RSLCnano system (Thermo Fisher Scientific, Waltham, MA, USA). The peptides were loaded onto a trap column (PepMap 100 mm × 2 cm; Thermo Fisher Scientific, Waltham, MA, USA) and separated on an analytical column (EasySpray, 75 mm × 50 cm; Thermo Fisher Scientific, Waltham, MA). Mobile phase A and mobile phase B were composed of 0.2% formic acid in 98% water/2% acetonitrile and 0.2% formic acid in 80% acetonitrile/10% isopropanol/10% water, respectively. Peptides were separated over 120 min from 3 to 40% of mobile phase B at a flow rate of 300 nl/min. The mass spectrometer was operated in a data-dependent mode with a cycle time of 3 sec. MS1 scans (375–1700 m/z) were acquired with an orbitrap resolution of 120,000, automatic gain control (AGC) of 200% and maximum injection time of 50 ms. Precursor ions were isolated in the quadrupole with an isolation width of 0.7 m/z and fragmented with high-energy collision-induced dissociation (HCD) at a normalized collision energy of 34%. MS/MS scans were acquired with an orbitrap resolution of 60,000, AGC of 200%, and a maximum injection time of 118 ms.

The raw data were searched against the SwissProt human database (release 2020_03, 20304 entries) using Sequest HT in Proteome Discover (ver2.5. Thermo Scientific, Waltham, MA, USA) with the parameter settings at 10 ppm for the precursor mass tolerance, 0.6 Da for the fragment mass tolerance and full trypsin specificity. Carbamidomethylation of cysteines and TMT label (+229.163) of peptide N-termini and lysines were set as static modifications and oxidation of methionine and acetylation of protein N-termini were considered as dynamic modifications during the database search. Identification was based on a 1% false discovery rate threshold applied at peptide and protein levels. TMT reporter ion quantitation was based on PSMs with a co-isolation threshold of <50% with an average reporter signal-to-noise threshold of at least 10. The abundance values were normalized to the same total peptide amount per channel and scaled so that the average abundance per protein and peptide was 100.

**ECM deposition assay**. Cells were plated in clear-bottom 96-well plates. After attachment, cells were incubated with CM generated from control or ERG-silenced cells. After 72 h, cells were fixed in 3.7% formalin (Sigma-Aldrich), blocked with Odyssey Blocking Buffer (LI-COR Biosciences, Lincoln, NE, USA) for 60 min, and incubated with anti Collagen I (NB600-408, Novus Biologicals, Littleton, CO, USA) and Fibronectin (sc-9068, Santa Cruz Biotechnology, Santa Cruz, CA, USA) antibodies (1:200 dilution in blocking buffer, overnight, 4 °C). The day after, the cells were stained with secondary goat anti-mouse IgG IRDye™ 800 antibody and goat anti-rabbit IgG IRDye™ 680 antibody (1:750 dilution, 45 min, RT). The microplates were scanned with the Odyssey CLx Infrared Imaging System (LI-COR Biosciences, Lincoln, NE, USA), and the integrated fluorescence intensities were acquired using the software provided with the imager station (Odyssey Software Version 3.0, LI-COR Biosciences).

**Protein extraction and western blot analysis**. For protein analysis, cells were lysed with RIPA buffer (Thermo Fisher Scientific, Waltham, MA, USA) supplemented with Halt™ protease & phosphatase inhibitor (Thermo Fisher Scientific, Waltham, MA, USA). Protein concentration was quantified with the Pierce BCA Protein Assay kit (Thermo Fisher Scientific, Waltham, MA, USA). Protein lysates were separated by electrophoresis, transferred onto PVDF membranes, and incubated overnight at 4° with primary antibodies: Collagen I (NB120-6586, Novus Biologicals Inc., Centennial, CO, USA, 1:1000 dilution), αSMA (ab5694, Abcam, Cambridge, MA, USA, 1:1000 dilution) and GAPDH (2118, Cell Signaling, Danvers, MA, USA, 1:1000 dilution). Blots were then washed and incubated with appropriate antibodies for 1 h at room temperature. Bands were visualized by using ChemiDoc Imaging System (Bio-Rad, Hercules, CA, USA), according to the manufacturer's protocol.

**Statistical analysis**. Individual data points are shown in all plots and represent data from independent mice, cells, or biological replicates from cell culture experiments. Depending on the group size, normality distribution was assessed with D'Agostino-Pearson omnibus, Shapiro–Wilk, or Kolmogorov–Smirnov normality tests. Variables are summarized as mean and SD, with a statistical comparison between two groups performed using Student's $t$-test and a comparison of more than two groups performed using a one-way analysis of variance (followed by Tukey's post hoc test). PCA graphs were created using a web tool for visualizing the clustering of multivariate data (https://biit.cs.ut.ee/clustvis/). Analysis using public R packages and general file parsing was performed using R v4.0.3 (https://www.R-project.org/). All analyses, plots, and heatmaps were generated using GraphPad Prism 8.4.3 (La Jolla, CA, USA) or NGSPLOT[79], with statistical significance defined as $p < 0.05$.

**Reporting summary**. Further information on research design is available in the Nature Research Reporting Summary linked to this article.

## Data availability

Raw and analyzed ATAC-seq, RNA-seq, and scRNA-seq data generated in this study are available through the Gene Expression Omnibus under the GEO accession numbers GSE177055, GSE181508, and GSE187333, respectively. The mass spectrometry proteomics data have been deposited to the ProteomeXchange Consortium via the PRIDE partner repository with the dataset identifier PXD033847 and https://doi.org/10.6019/PXD033847. All data supporting the findings described in the manuscript are available in the article and in the supplementary information, and from the corresponding author upon reasonable request. Source data are provided with this paper.

## Code availability

All in-house scripts are previously described[70] or are publicly available through zenodo (https://zenodo.org/record/6543252#.Yn6uM-jMI2w).

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

## Acknowledgements

We thank Dr. Derek Radisky (Mayo Clinic) for kindly providing Col1α1-GFP transgenic mice, Mrunal K. Dehankar (Mayo Clinic), and Tianmu Hu (Boston University) for supporting bioinformatics analysis, Dr. Tamas Ordog and Dr. Jeong-Heon Lee (Mayo Clinic) for thoughtful insights on processing samples for ATAC-seq, and Dr. Akhilesh Pandey, Dr. Cristine M. Charlesworth, and Benjamin C. Brown (Mayo Clinic) for supporting proteomic analysis. We gratefully acknowledge the support of this work by the National Institutes of Health (NIH) grants HL142596 (G.L.), HL158733 (G.L.), HL150638 (M.T.), AR060780 (M.T.), HL092961 (D.J.T.), HL105355 (P.A.L. and D.L.J.), HL007035 (T.X.P.), the American Lung Association Dalsemer Award (N.C.), and the Boehringer Ingelheim Discovery Award IPF/ILD (N.C.).

## Author contributions

N.C. and G.L. designed the study, N.C., J.L., J.G., J.A.M., G.M., and T.Y. performed the experiments, N.C., G.L., D.L.J., P.A.L., T.X.P., C.A.O., and A.V.B. analyzed the data. The manuscript was drafted by N.C. and G.L. and revised by M.T., D.J.T., R.F.N., and S.K.H. All Authors participated in manuscript preparation and provided final approval of the submitted work.

## Competing interests

The authors declare no competing interests.
