## [Peer Review File · Nature Communications]

Reviewers' Comments:

Reviewer #1:

Remarks to the Author:

This very interesting manuscript focuses on the effect of ageing and injury on chromatin accessibility and gene expression in lung endothelial cells; they show that the transcription factor ERG is a key regulator of lung EC function and ability to repair; they implicate ERG in the dysfunction associated with ageing and injury and propose these mechanisms as model for the human disease IPF.

The role of ERG in determining endothelial lineage and maintaining homeostasis has been shown in multiple studies; here Caporarello et al propose a link between loss of ERG, lung EC injury and EC ageing. The approach involves ex vivo genomic studies on lung microvascular endothelium from young and aged mice exposed to Bleomycin, in vitro mechanism studies on the pro-fibrotic effect of ERG ko, single cell RNASeq mapping of lungs from endothelial-specific inducible ERG-ko mice and whole lung scRNA-seq from human IPF tissues. Through these last experiments, the Authors identify a role for ERG in the specification of gCap cells (a subset of lung EC involved in alveolar vascular repair) and describe a molecular connection between loss of ERG and IPF.

The study is well conducted, clearly presented, mostly novel and of great interest to multiple fields, from the organotypic control of endothelial gene expression to the translational relevance to IPF. The evidence about ageing resulting in different chromatin accessibility in endothelial cells a mouse model is convincing, and in line with evidence from other cell lineages, but little is known about it in EC. The evidence for the role of ERG in regulating lung EC function and fibrosis has been shown before, but the Authors bring novelty to this by analysing the secretome of ERG-deficient EC. The link between ERG and ageing is indirect and thus still speculative. The story becomes a bit disconnected when the Authors attempt to link defective response to injury in aged mice (reflected/caused by defective chromatin rearrangement), decreased ERG function and gCap cells. The hypothesis is fascinating, but there are several gaps in the story. Is the chromatin accessibility phenotype due to lack of or decreased ERG function in aged mice exposed to injury? What is the relationship between the differentially accessible chromatin sites in aged mice and the genes regulated by ERG? These and some other points should be addressed to support the model. A graphical model that summarises the hypothesis would be useful.

Specific Comments:

1. Fig 1C: the analysis shows a clear difference in chromatin accessibility between aged vs young mice. For the other groups, injured young and injured aged ECs were compared to young sham; so both Bleomycin responses were compared to young mice. This may mask differences due to aging rather than treatment and ageing. The 4th comparison - age-matched - should be included in the analysis, unless the Authors can provide a satisfactory explanation for the rationale for leaving it out. The density curves in the right panel are useful visual tools to capture they overall observation; they should be more clearly labelled to define each condition.

2. Motif analysis of differentially accessible sites across the different groups points to ERG as a key regulator of the response to bleomycin. The authors conclude that "aging affects the ability of ERG to engage chromatin remodelling and exert its transcriptional function to effectively respond to lung injury." This is intriguing. But the comparison between Aged mice sham vs Aged mice with bleomycin should be included in the analysis, in order to determine which differences are specifically due to the injury and which are due to young vs aged.

3. A key question which is not addressed is why should ERG lose its ability "to engage chromatin remodelling and exert its transcriptional function to effectively respond to lung injury ", as stated in the paper? Are ERG levels reduced in Aged mice? Or is it ERG's post-translational modification, such as phosphorylation, which is differentially regulated in Aged mice? We know that Erg levels are downregulated by inflammation. It would be important to at least report ERG's protein levels in the various conditions to support the statement above. Also, this point should be addressed in the discussion.

4. The Authors carry out both RNASeq and ATACseq on EC from the groups of interest; this is a very valuable approach with potential to understand the relationship between chromatin regulation and the phenotype of the cells. What is the relationship between the results from the RNASeq analysis and the ATACSeq data? One would expect a degree of correlation. The Authors should carry out detailed comparison of the datasets to identify the pathways which connect the

chromatin accessibility side of the story with the gene expression phenotype. The focus on ERG comes from their interest in chromatin accessibility; hence the data analysis should support this.

5. Fig 5: the results in this figure aim to demonstrate that 1) Erg regulates angiogenesis and vascular assembly, 2) loss of ERG is pro-fibrotic and 3) loss of ERG recapitulates the gene expression signature observed in lung ECs from aged mice.

The first point is already established by previous publications. The role of ERG in vascular assembly and angiogenesis has been already described in detail, as well as ERG's role in endothelial junction formation and cell-cell adhesion. Thus, the findings in Fig 5A-B and described in lines 320-325 should be presented as valuable confirmation in this EC subtype of previous findings.

7. The pro-fibrotic effects of ERG loss in EC have been described in other organs (the Authors refer to this elsewhere in their paper), but their experiments with Conditioned medium from ERG-depleted EC on fibroblasts, including the profiling of the cell's secretome, are novel and compelling.

8. The third point, on the relationship between ERG targets and genes dysregulated in EC during ageing mice, is only made indirectly by the Authors and is based on qPCR profiles. The argument that ERG dysfunction drives the aged mice phenotype requires, at the molecular level, a comparison between Transcription profiles of ERG-deficient EC and aged EC in the lung. If the Authors have not generated or do not have access to RNASeq from ERG-deficient lung EC, they should consider using existing datasets of ERG-deficient EC from other sources, take into account endothelial heterogeneity and focus on the shared targets, to compare these with aged EC and provide evidence to support their point. It may be possible also to use the lung scRNASeq generated from the inducible ERG ko mice, although the bioinformatic comparison between sc and bulk RNASeq datasets is challenging and not ideal.

9. In fig 6, the inducible endothelial ERG KO mice are exposed to injury, and show a more severe phenotype compared to controls, including signs similar to those observed in Aged mice (reduced vascular remodelling and fibrosis). Investigating in vivo some molecular targets identified earlier in the study as shared between ERG deficiency and ageing would complete this experiment and support the argument.

10. In the final experiment, the Authors carry out scRNASeq on ERG deficient lungs, and process the data focussing entirely on EC. Within the different EC types, they highlight gCap cells, a recently described subset of lung endothelial progenitors linked to vascular regeneration in the alveoli. They find that gCap cells are decreased in ERG-KO lungs and therefore speculate that ERG controls gCap cell identity and through these vascular regeneration in the lung. These are novel and interesting data, which support this hypothesis but do not convincingly prove it. Rescue experiments, lineage tracing studies and more detailed characterisation will be required in the future to confirm the hypothesis; thus the text needs to reflect the preliminary nature of this observation. The scRNASeq is also likely to provide very exciting novel information on the indirect effects of endothelial ERG deletion on other cells; given the focus on fibrosis in this study, it would be really interesting to know what phenotypic changes occur in fibroblasts and other cells that contribute to fibrosis. The Authors have asked these questions with the in vitro experiments in Fig 5, using endothelial CM on fibroblasts; using the scRNASeq data, they could confirm their finding in vivo, which would be very powerful.

Minor comments

1. Fig 1A: the PCA shows a considerable spread of the Aged-Bleo samples. Do the Authors have any comments regarding this finding, and how this might possibly have affected their subsequent data analysis?

2. Fig 3A: PCA analysis of RNASeq samples does not really show a good separation between samples, as stated in the paper. More in depth QC of the samples addressing this point should be reported in supplementary, and the statement on line 224 rephrased.

3. The scRNASeq was carried out on whole lungs 90 days after tamoxifen injections. This is an unusually long time, and different from the timepoint chosen for the bleomycin experiments (day 45 post-bleomycin delivery). Could the authors clarify why they chose this timepoint, whether they repeated TX injections and if not whether they were able to confirm that ERG deletion was stable

throughout?

Reviewer #2:

Remarks to the Author:

This manuscript, written by Caporarello, describes the involvement of a transcription ERG in aging-related lung repair and potential pathogenesis of persistent lung fibrosis, such as idiopathic pulmonary fibrosis (IPF). The authors first compared the epigenomic patterns in lung endothelial cells (ECs) between young and aged mice using ATAC seq. They found that the aged mice generally show a close epigenomic pattern. They identified binding sites of a transcription factor, ERG, were resided in those differentially regulated regions. They further examined the epignomic changes in response to a bleomycin stimulation, which is a model of lung injury. They found that ERG may play the pivotal role in the regulation of a series of genes which are involved in lung capillary homeostasis and repair following the injury. Using an in vitro experiment in a human lung microvascular endothelial cells (HLMECs), they conducted siRNA and recapitulated the enhanced fibroblast activation and collagen deposition which they observed from the mouse analyses. They also conducted a single cell RNA sequencing (scRNA seq) analysis of ERG deficient mice. They found that the population of general capillary (gCap) ECs. Finally, they utilized public scRNA seq data of human IPFs and found that the reduced numbers of gCap ECs and vascular progenitor cells are also represented there. Overall, I fully appreciate the authors efforts to deepen our understandings for the relation between aging and etiology of IPF. However, I have substantial remaining concerns as to whether the presented data collectively lead the same conclusion. Even though the collected material should potentially include all the necessary information, some key analyses are missing for every part.

Major points:

1. Given the overall population changes in lung ECs was identified from the scRNA seq analysis, it is essential to further characterize which cell types are responsible for the observed epigenome changes. The observed gene expression changes should be also re-considered, perhaps by making use of a deconvolution analysis. Recently, an analytical method for the simultaneous single cell analysis of ATAC seq and RNA seq (such as scMultiome) has been commercialized. Such an analysis would give an essential information to re-evaluate the results of Figs 1-4.
2. Analyses for the scRNA seq data of human IPF is too shallow. The consistency and the inconsistency against the mouse model should be more intensively examined. A particular focus should be put on the status of ERG activity, depending on the patients. Also, I wonder if there are any clinical symptoms associated with the molecular status of the ECs of the patients?
3. Given that differentially accessible sites in promoters are relatively rare, contribution of ERG for the overall etiology of IPF should be carefully considered.

Minor points:

4. I wonder when the function of ERG is deteriorated with aging to eventually give an epigenomic condition which is vulnerable to IPF. Increasing data points would answer the question.
5. When and how does ERG realize the most important role after the injury. Further detailed data collection and analyses should be needed before during the periods of inflammatory responses, lung vascular repair and the establishment of eventual fibrosis. For the fibrosis, the data points are too limited to discuss the "persistence".
6. The authors have already identified Nos3 as a key player for persistent lung fibrosis (reference 17). I wonder Nos3 is the major target of ERG? In other words, the function of ERG is mostly in proper induction of Nos3? Further careful analyses should be needed for additional possible roles of ERG.
7. Degree of differential accessibility and the gene expression levels should be mutually compared.

8. Fig. 6: Essential controls seem missing from ERG CKO (Bleomycin minus). This control is important because it is possible that ERG CKO may have developed predisposed lesions.
9. Figure 6C: The timing for the conditional KO of ERG is not clear to me. If this is a model of an aged lung, the KO should be induced before the Bleomycin treatment?
10. The first sentence of the abstract is misleading. Strictly speaking, I do not think this is the paper on human IPF.
11. Discussion section is, at least to some extent, repeating the descriptions in the former sections, thus, could be shortened.

Reviewer #3:

Remarks to the Author:

This manuscript from Ligresti and colleagues reported that dysregulation of ERG impairs lung vascular repair and promotes fibrosis in aging mice. Although the proposed data would be interesting, the study lacks a logical connection and clear mechanistic insights. The data analysis should be improved and more experimental data are required for demonstrate the mechanism about how ERG mediates vascular repair in the aging mice.

Here are some important concerns to highlight:

1. The sequencing data analysis should be improved. The data in Fig.1A does not support the authors' statement that "extensive differences in chromatin accessibility, with samples clustering into different groups based on age and responses to bleomycin injury". In fact, the Sham and Bleo group mixed together for the 3 of 5 replicates in the aged mice, representing the data are not so qualified.
2. For Fig.1B, the author cannot just use squares for regions with divergent chromatin accessibility, which is quite inaccurate and unacceptable. The authors should divide these regions into different groups according to the accessibility change patterns and then plot them separately.
3. The authors have performed ATAC-seq and RNA-seq, but analyzed the data separately, it would be better to perform joint analysis combined ATAC-seq with RNA-seq. For ATAC-seq, taking Fig.1F as example, the author need to check if those more/less accessible loci are relative to their proximal gene up/down regulation in RNA-seq. Besides, it would be better to show chromatin accessibility and gene expression with the genomic plot by IGV etc instead of only the gene names in case of any misinterpretation because ATAC-seq peak are mostly located at distal regulatory elements such as enhancers but not gene, and each gene would include several ATAC-seq peaks.
4. For Fig.2, it is really difficult to follow the logic of such analysis. It would be better if the authors can perform the comparative analysis that how many sites that are accessible in the young mice but closed in the old mice, and also take the reverse part (open in the old mice but closed in the young). Similarly, loci groups relative to Bleo response can be summarized and then motif enrichment analysis should be performed for those sub-clusters of ATAC-seq peaks.
5. The ERG ChIP-seq data are required for the validation of the potential target genes of Fig. 2J-L.
6. Again, the Fig3A does not support the authors' claim that "PCA analysis showed that samples formed distinct clusters, with ECs of injured and uninjured lungs sharing the least similarities", the samples can't be sperate correctly by both PC1 and PC2, and showing that significant difference between replicates, which also simply reflected by Fig.3D-F. We have some concerns about the data quality and consistency.
7. For Fig.3B-C, how did the author perform that analysis? Detailed statement about the analysis pipeline and the corresponding citations should be included in the method part.
8. Fig.3H, the KD efficiency need to be shown.
9. Line 248 & Line 394, the data need to be shown.
10. Fig.7B-D, the color scheme makes the figure difficult to read. Moreover, the expression of all the marker genes for each cell type should be shown in the supplementary figures.
11. Lack of the molecular mechanism makes the manuscript incomplete, it is still unclear why ERG are important for capillary homeostasis, and it is also unclear whether the gCap cells contribute to injury repair. An increased focus on ERG coupled mechanisms may improve the manuscript.

Minor issues:

1. For Fig.1C, the "open/close" should be "more/less accessible" for consistency and to avoid misleading.
2. LINE153, Homer need a citation.

Reviewer #4:

Remarks to the Author:

The manuscript entitled "Dysregulation of the transcription factor ERG in aging impairs lung vascular repair and promotes persistent fibrosis" by Caporarello and collaborators describe the involvement of ERG signaling in the homeostasis and response to repair of lung vasculature. The topic of manuscript is exiting and highly relevant. However, some conceptual and experimental designs raise major concerns that limit my enthusiasm for this manuscript.

Major concerns:

1) The authors open with this statement: "Here, we employed epigenetic and transcriptional analysis using ATAC-seq and RNA-seq of freshly isolated lung endothelial cells (ECs) from young and aged mice to identify endothelial programs during the resolution or progression of bleomycin-induced lung fibrosis." However, the author used 30 days post bleomycin instillation as their main endpoint. In their previous publication (2020 Aging Cell -figure1c-d), the levels of hydroxyproline in the lung of the young mice at day 30 is as high (if not higher) than the older mice. Only at day 75 post-installation they report a difference. That will be the correct time window to address any changes in resolution, since (base on their own data) day 30 is still a highly fibrotic. Further, the authors should compare cell populations and gene expression in endothelial cells from resolved fibrosis to peak of fibrosis (i.e. both animals subjected to bleomycin application).

2) ERG cKO studies (Figure 6): The design of this animal experiments is lacking the appropriate controls. Tamoxifen has some anti-angiogenic effects, so the proper control should be Erg fl/fl mice with tamoxifen and PBS instillation (instead of only PBS). This would be a more comparable uninjured WT model. In addition, the authors do not report any fibrotic-related data on the changes (or absence of changes) in the ERG cKO when uninjured. Is the vascular leakage promoting fibrotic changes in the uninjured cKO mice? Also, is this leakage affecting different size vessels differently?

3) Figure 7: In this study a third different timepoint (90 days after recombination) is used. None of these data then show changes than can be used to complement Figure 6. In addition, at this point of the manuscript the authors bring up the different sources of the ECs in the lung (all previous studies were done as if there were one single population). If the authors want to venture in the understand the changes in aCap and gCap with aging (and in the context of the ERG KO), this single cell study should be compared with aged mice (both WT and KO).

4) Methods: In the description of the mouse model, detailed information of the type of anesthesia and concentration/regimen of the tamoxifen injection should be added. Equally important should be to describe in detail the experimental set-up of the cell culture studies (i.e. concentration of siRNA, time of transfection, time of harvest, etc). These additions will ensure the reproducibility of any of the experiments.

Minor concerns:

Figure 1: Despite the labeling, it is not evident that panel A & B represent data from the four experimental groups and the subsequent panel are just the 3 comparisons between groups. Seems confusing upon first reading and limits the excitement of the data.

Figure 3: The use of blue as a group identificatory "aged sham" on panel A plus as part of the colored scale in the heatmap is confusing. Also, color scheme used in panel G is not carried equally for all the TSS analysis.

Figure 4: The color scheme is not consistent between the same groups on panels I, J and K.

Figure 5: Microscopy images in this figure are lacking proper scale bars. In addition, panel H does not add any results or information not already mentioned in the method section. Similarly, the schemes in panels A and D seems superfluous and are missing key information such as harvesting time or dilution ration of the conditioned media in fresh media (if any).

Figure 6: Except for panel A, all other microscopy images in this figure are lacking proper scale bars.

Methods: catalog numbers for the commercially available mice are missing. In addition, clone identification (or catalog number) should be added to the antibodies using in this study.

We thank the reviewers for their thoughtful and constructive critiques. Below are the reviewers' comments reported verbatim and our point-by-point responses **shown in bold**.

Reviewer #1 (Remarks to the Author):

This very interesting manuscript focuses on the effect of ageing and injury on chromatin accessibility and gene expression in lung endothelial cells; they show that the transcription factor ERG is a key regulator of lung EC function and ability to repair; they implicate ERG in the dysfunction associated with ageing and injury and propose these mechanisms as model for the human disease IPF.

The role of ERG in determining endothelial lineage and maintaining homeostasis has been shown in multiple studies; here Caporarello et al propose a link between loss of ERG, lung EC injury and EC ageing. The approach involves ex vivo genomic studies on lung microvascular endothelium from young and aged mice exposed to Bleomycin, in vitro mechanism studies on the pro-fibrotic effect of ERG ko, single cell RNASeq mapping of lungs from endothelial-specific inducible ERG-ko mice and whole lung scRNA-seq from human IPF tissues. Through these last experiments, the Authors identify a role for ERG in the specification of gCap cells (a subset of lung EC involved in alveolar vascular repair) and describe a molecular connection between loss of ERG and IPF.

The study is well conducted, clearly presented, mostly novel and of great interest to multiple fields, from the organotypic control of endothelial gene expression to the translational relevance to IPF. The evidence about ageing resulting in different chromatin accessibility in endothelial cells a mouse model is convincing, and in line with evidence from other cell lineages, but little is known about it in EC. The evidence for the role of ERG in regulating lung EC function and fibrosis has been shown before, but the Authors bring novelty to this by analysing the secretome of ERG-deficient EC. The link between ERG and ageing is indirect and thus still speculative. The story becomes a bit disconnected when the Authors attempt to link defective response to injury in aged mice (reflected/caused by defective chromatin rearrangement), decreased ERG function and gCap cells. The hypothesis is fascinating, but there are several gaps in the story. Is the chromatin accessibility phenotype due to lack of or decreased ERG function in aged mice exposed to injury? What is the relationship between the differentially accessible chromatin sites in aged mice and the genes regulated by ERG? These and some other points should be addressed to support the model. A graphical model that summarises the hypothesis would be useful.

Specific Comments:

1. Fig 1C: the analysis shows a clear difference in chromatin accessibility between aged vs young mice. For the other groups, injured young and injured aged ECs were compared to young sham; so both Bleomycin responses were compared to young mice. This may mask differences due to aging rather than treatment and ageing. The 4th comparison - age-matched - should be included in the analysis, unless the Authors can provide a satisfactory explanation for the rationale for leaving it out. The density curves in the right panel are useful visual tools to capture they overall observation; they should be more clearly labelled to define each condition.

Response: We decided to not include the comparison between aged sham ECs vs aged injured ECs as this analysis resulted in a very low number (88) of differentially accessible chromatin regions

due to the variability of the samples in the aged groups. With such low number of differentially accessible sites, the motif analysis we performed did not generate meaningful outcomes. This finding may be related to inherent inter-individual heterogeneity in chromatin accessibility associated with aging. This interpretation is corroborated by a recent study showing increased transcriptional noise in mouse lung cells with aging, due to dysregulated epigenetic control (1). In our study, while young ECs (sham and injured) clustered quite uniformly in our PCA analysis (Fig. 1A), aged ECs (sham and injured), exhibited a more scattered distribution, suggesting that aged mouse ECs produce different responses to bleomycin likely due to their underlying more heterogeneous chromatin accessibility. Thus, we decided to use young sham ECs as reference group for both young and aged injured ECs. While we agree that the age-matched comparison would provide a more inclusive evaluation of the contribution of vascular aging to lung injury responses, our data obtained with the sham ECs control group still provide compelling information on differentially accessible chromatin sites in aged ECs relative to young ones. For example, the chromatin accessibility of gCap markers genes, such as *Aplnr*, *Cd93*, *Kit* and *Plvap*, were diminished in both aged sham and aged injured ECs compared to young ECs. We feel that reporting the reason why the age-matched comparison was omitted from the study would be helpful for the readers, and so we have included this information in the results.

Regarding the density curves, we have modified the labels to enhance clarity. A schematic summarizing our hypothesis is provided in Supplementary Figure 8.

2. Motif analysis of differentially accessible sites across the different groups points to ERG as a key regulator of the response to bleomycin. The authors conclude that “aging affects the ability of ERG to engage chromatin remodeling and exert its transcriptional function to effectively respond to lung injury.” This is intriguing. But the comparison between Aged mice sham vs Aged mice with bleomycin should be included in the analysis, in order to determine which differences are specifically due to the injury and which are due to young vs aged.

Response: Our analysis between young and aged lung ECs showed ERG motif enrichment in most of the non-accessible chromatin regions in aged lung ECs. In addition, our newly performed comparative analysis between RNA-seq and ATAC-seq revealed that the large majority of non-accessible chromatin regions containing ERG motifs were located near the promoters of angiogenic genes whose expression was impaired in injured aged lung ECs and have been previously implicated in ERG-mediated responses. Though we agree that a direct comparison between uninjured aged EC and injured aged ECs would provide a more comprehensive understanding of the role of ERG-mediated lung vascular responses in aging, an analysis at single gene level still provided critical information that led us to conclude that ERG responses to injury are impaired with aging. For example, we found no difference in *Lama4*, *Hspg2*, and *Bmp2* chromatin accessibility with aging, however, upon injury the chromatin accessibility of these ERG target genes was increased in young lung ECs but not in aged lung ECs (Fig. 4L and M). These findings led us to hypothesize that ERG-mediated chromatin remodeling plays a role in lung vascular responses to bleomycin and this function was impaired in aging. On the other hand, we also found that endothelial identity genes containing ERG motifs, such as *Aplnr* (encoding for Aplin receptor), exhibited reduced chromatin accessibility in aged lung ECs compared to young lung ECs

even in absence of injury, suggesting that ERG-mediated lung vascular homeostasis may be also compromised with aging.

3. A key question which is not addressed is why should ERG lose its ability “to engage chromatin remodeling and exert its transcriptional function to effectively respond to lung injury “, as started in the paper? Are ERG levels reduced in Aged mice? Or is it ERG’s post-translational modification, such as phosphorylation, which is differentially regulated in Aged mice? We know that Erg levels are downregulated by inflammation. It would be important to at least report ERG’s protein levels in the various conditions to support the statement above. Also, this point should be addressed in the discussion.

Response: We agree with the reviewer that this point was not clearly discussed in the original manuscript. We have therefore edited the discussion to include the following important considerations regarding possible mechanisms underlying ERG contribution to lung repair and vascular aging.

As stated by the reviewer, previous studies have shown that inflammatory stimuli inhibited ERG expression in aortic- and umbilical cord -derived ECs (2, 3), and these data were confirmed in our laboratory using lung-derived ECs (data not shown). However, the RNA-seq analysis we carried out on isolated mouse lung ECs showed no reduction in ERG expression in either injured or uninjured aged ECs relative to young ECs (we actually observed a modest but significant increase in Erg gene expression equally in both injured young ECs and injured aged ECs relative to uninjured young ECs). Interestingly, our ATAC-seq analysis revealed that numerous less accessible chromatin regions in aged lung ECs, compared to young lung ECs, contained ERG binding sites leading us to hypothesize that a non-permissive chromatin state or an abnormal ERG signaling in aging may affect its chromatin binding.

In support of the first hypothesis (non-permissive chromatin state), previous studies including one by our collaborator and co-author Dr. Trojanowska reported that inflammatory cytokines, such as $IFN\alpha$, IL1 or $TNF\alpha$, can alter endothelial chromatin to prevent ERG binding leading to endothelial inflammation (4, 5). Another study demonstrated that ERG and the transcription factor NF κ B can compete with each other in ECs for chromatin binding and transcriptional activity (6), suggesting that aging-associated endothelial inflammation may ultimately affect ERG chromatin interaction.

In support of the abnormal ERG signaling, the inability of ERG to interact with chromatin-modifying enzymes in aged lung ECs may potentially affect endothelial chromatin remodeling, histone decoration, and gene transcription. For instance, a recent study showed that loss of ERG in human ECs derived from the umbilical cord (HUVECs) affected histone mark decoration, specifically H3K27ac, at enhancers, and this alteration was due to a reduced occupancy of the acetyltransferase p300 at ERG target genes (7), suggesting that that ERG-mediated chromatin remodeling *via* epigenetic regulators may be compromised with aging.

Furthermore, a recent study reported that ERG chromatin binding is dependent on its phosphorylation status (8). In this paper, it was shown that the kinase Akt, which is downstream the Angiotensin-1/TEK axis, is directly responsible for ERG phosphorylation, and loss of this modification impaired ERG binding to distal regulatory regions. Intriguingly, our multi-omics analysis revealed that chromatin accessibility and expression of *Tek* gene were strongly reduced in

aged ECs compared to young ECs, suggesting that aberrant TEK signaling in aging may compromise ERG activation and chromatin binding, thereby affecting gene transcription. Also, Angiopoietin-2, an antagonist of TEK signaling, was found to be elevated in the plasma of patients with idiopathic pulmonary fibrosis (9), further suggesting that reduced TEK signaling in IPF may lead to dysregulated ERG activity and vascular dysfunction.

Future experiments will be required to demonstrate which of these mechanisms are involved in the impaired ERG function with aging.

4. The Authors carry out both RNASeq and ATACseq on EC from the groups of interest; this is a very valuable approach with potential to understand the relationship between chromatin regulation and the phenotype of the cells. What is the relationship between the results from the RNASeq analysis and the ATAC-Seq data? One would expect a degree of correlation. The Authors should carry out detailed comparison of the datasets to identify the pathways which connect the chromatin accessibility side of the story with the gene expression phenotype. The focus on ERG comes from their interest in chromatin accessibility; hence the data analysis should support this.

Response: We have carried out a comparative analysis between RNA-seq and ATAC-seq (for both uninjured ECs, Figure 3G, and injured ECs, Figure 4I and J) and found that gene expression moderately correlated with ATAC-seq. For example, in Fig.3D we showed that innate immunity genes were upregulated in aged lung EC compared to young lung ECs, however, the level of chromatin accessibility was similar between these two groups. Similarly, Fig.4F shows increased expression of genes implicated in cell proliferation in both injured young ECs and aged lung ECs relative to uninjured young lung ECs, whereas chromatin accessibility levels were not different (Fig.4K) between these groups. These findings demonstrated that not all transcriptional changes we observed in lung EC (aging- or injury-mediated) resulted from alterations in chromatin accessibility. Strikingly, we found that numerous genes that positively correlated with chromatin accessibility were angiogenesis-associated genes and endothelial identity genes, and the large majority of them contained ERG motifs in their regulatory regions. We have included these new data in Fig.3G and in Fig.4I and J and edited the text to reflect these changes.

5. Fig 5: the results in this figure aim to demonstrate that 1) Erg regulates angiogenesis and vascular assembly, 2) loss of ERG is pro-fibrotic and 3) loss of ERG recapitulates the gene expression signature observed in lung ECs from aged mice. The first point is already established by previous publications. The role of ERG in vascular assembly and angiogenesis has been already described in detail, as well as ERG's role in endothelial junction formation and cell-cell adhesion. Thus, the findings in Fig 5A-B and described in lines 320-325 should be presented as valuable confirmation in this EC subtype of previous findings.

Response: We agree with the reviewer that ERG contribution to angiogenesis has already been established. However, we decided to include these findings in the manuscript to confirm ERG function specifically in lung ECs. We have edited the text to better emphasize the confirmatory nature of these experiments in our system.

7. The pro-fibrotic effects of ERG loss in EC have been described in other organs (the Authors refer to this elsewhere in their paper), but their experiments with Conditioned medium from ERG-depleted EC on fibroblasts, including the profiling of the cell's secretome, are novel and compelling.

Response: We thank the reviewer for the positive feedback.

8. The third point, on the relationship between ERG targets and genes dysregulated in EC during ageing mice, is only made indirectly by the Authors and is based on qPCR profiles. The argument that ERG dysfunction drives the aged mice phenotype requires, at the molecular level, a comparison between Transcription profiles of ERG-deficient EC and aged EC in the lung. If the Authors have not generated or do not have access to RNASeq from ERG-deficient lung EC, they should consider using existing datasets of ERG-deficient EC from other sources, take into account endothelial heterogeneity and focus on the shared targets, to compare these with aged EC and provide evidence to support their point. It may be possible also to use the lung scRNASeq generated from the inducible ERG ko mice, although the bioinformatic comparison between sc and bulk RNASeq datasets is challenging and not ideal.

Response: We are glad this important point was raised by the reviewer. Since we couldn't find publicly available RNA-seq data from ERG-deficient lung ECs, we have compared our bulk RNA-seq (Aged vs Young) with the scRNA-seq (ERG CKO vs WT) and generated new data that are now included in the updated Figure 8G. Interestingly, we found that roughly 20% of the genes whose expression was altered in ECs with aging were also dysregulated in ECs lacking ERG. Among them were numerous gCap EC marker genes as well as inflammatory genes, suggesting that dysfunctional ERG signaling in aging may drive capillary dysfunction and inflammation. To strengthen this premise and further connect ERG dysfunction to the phenotypic manifestations of aging, we have included new data demonstrating that loss of endothelial ERG leads to increased lung inflammation. Among the inflammatory genes whose expression was increased in the lungs of ERG CKO mice were those that have been implicated in aging and senescence, including *Ilf6*, *Cxcl10*, *Serpina3n* (new Figure 6E). In addition, FACS analysis and MPO staining revealed increased neutrophils recruitment in the lungs of ERG CKO mice compared to WT ones (new Figure 6C and D). Consistent with these observations, previous studies have demonstrated elevated neutrophil recruitment in multiple organs with aging (10). Altogether, these data, along with our multi-omics analysis reinforced the concept that dysfunctional endothelial ERG activity with aging may lead to progressive deterioration of organ function. We have edited the manuscript and the figures to include these new data.

9. In fig 6, the inducible endothelial ERG KO mice are exposed to injury, and show a more severe phenotype compared to controls, including signs similar to those observed in Aged mice (reduced vascular remodelling and fibrosis). Investigating in vivo some molecular targets identified earlier in the study as shared between ERG deficiency and ageing would complete this experiment and support the argument.

Response: We have previously demonstrated that gene expression of endothelial *NOS3*, which is a well know ERG target gene (11), was significantly reduced in aged mice following bleomycin challenge (12). Similarly to the unresolved lung fibrosis observed both in aged and in ERG CKO mice, loss of endothelial *NOS3* in young mice leads to more severe fibrosis following bleomycin challenge (12).

In addition, in Figure 6G of the original manuscript we showed that the increased lung fibrosis in ERG CKO mice following bleomycin challenge was accompanied by the reduced deposition of basement membrane proteins, including Collagen IV and Laminin IV. Intriguingly, the chromatin accessibility and the expression of genes encoding for these two proteins (*Col4a1* and *Lama4*)

were reduced in aged ECs following bleomycin injury, further suggesting that dysfunctional ERG signaling in aging may affect the expression of basement membrane proteins, which are key to vascular repair following injury.

10. In the final experiment, the Authors carry out scRNASeq on ERG deficient lungs, and process the data focussing entirely on EC. Within the different EC types, they highlight gCap cells, a recently described subset of lung endothelial progenitors linked to vascular regeneration in the alveoli. They find that gCap cells are decreased in ERG-KO lungs and therefore speculate that ERG controls gCap cell identity and through these vascular regeneration in the lung. These are novel and interesting data, which support this hypothesis but do not convincingly prove it. Rescue experiments, lineage tracing studies and more detailed characterization will be required in the future to confirm the hypothesis; thus the text needs to reflect the preliminary nature of this observation. The scRNASeq is also likely to provide very exciting novel information on the indirect effects of endothelial ERG deletion on other cells; given the focus on fibrosis in this study, it would be really interesting to know what phenotypic changes occur in fibroblasts and other cells that contribute to fibrosis. The Authors have asked these questions with the *in vitro* experiments in Fig 5, using endothelial CM on fibroblasts; using the scRNASeq data, they could confirm their finding *in vivo*, which would be very powerful.

Response: In the future, we are planning to carry out lineage tracing studies to trace the fate of gCap ECs lacking ERG. Our hypothesis is that ERG is essential for gCap ECs repair following lung injury, and that its function is compromised in aged gCap ECs. We have now edited the discussion section to emphasize the preliminary nature of our observations.

We are grateful to the reviewer for this comment which prompted us to carry out additional analysis of our ERG CKO scRNAseq dataset. We have now included in Supplementary Fig.7 new data showing transcriptional alterations occurring in lung cells previously implicated in fibrosis, including macrophages, neutrophils, and fibroblasts. Transcriptional analysis of neutrophils and macrophages from ERG CKO animals revealed increased expression of cell activation markers, including *S100a9*, *Ccl6* and *Mmp9*, in these immune cells compared to those from WT. Interestingly, analysis of fibroblasts from ERG CKO group revealed increased expression of genes implicated in fibrosis and TGF β signaling, including *Ltbp2*, *Acta2*, *Tagln* and *Tnc*, in these cells compared to those from WT mice, thus strengthening our *in vitro* analysis.

Minor comments

1. Fig 1A: the PCA shows a considerable spread of the Aged-Bleo samples. Do the Authors have any comments regarding this finding, and how this might possibly have affected their subsequent data analysis?

Response: We did observe some variability in the aged groups of ECs (both uninjured and injured). This phenomenon may be related to inherent inter-individual heterogeneity in chromatin accessibility associated with aging. To alleviate potential concerns related to the quality of the DNA sequencing we have assessed the fraction of reads in peaks (FRIP) for each sample, which is defined as the fraction of mapped reads that fall into a peak and is often used as a measure of sequencing quality. Based on the ENCODE metrics, the ideal FRIP value is >0.3. For our ATAC-seq

analysis, all FRIPs were above 0.3. This additional information has been included in the method section of the revised paper.

2. Fig 3A: PCA analysis of RNASeq samples does not really show a good separation between samples, as stated in the paper. More in depth QC of the samples addressing this point should be reported in supplementary, and the statement on line 224 rephrased.

Response: We have checked the quality of the samples based on the RNA integrity number, RIN. A good quality RNA should have a RIN >7. Except for one sample which have a RIN=7.9, all the RNA samples had RINs between 8 and 10, confirming the good quality of these samples. We have included this information in the method section and rephrased the statement on line 224 of the original manuscript in the revised version of the paper.

The scRNA-Seq was carried out on whole lungs 90 days after tamoxifen injections. This is an unusually long time, and different from the timepoint chosen for the bleomycin experiments (day 45 post-bleomycin delivery). Could the authors clarify why they chose this timepoint, whether they repeated TX injections and if not whether they were able to confirm that ERG deletion was stable throughout?

Response: The decision of carrying out the scRNA-seq at 90 days post tamoxifen delivery (daily administrations for 5 consecutive days) was based on previous experiments (included now in the revised version in Figure 6F and G) showing perivascular remodeling around large as well as small vessels without noticeable fibrosis in ERG CKO mice at day 45 post tamoxifen delivery compared to WT mice (in absence of injury). Given that collagen and other ECM proteins slowly accumulate in the lung interstitium during pulmonary fibrosis, we reasoned that ECM protein accumulation would be more established/severe after a prolonged period of time. Intriguingly, besides the reduction of gCap ECs, our scRNA-seq also revealed that bystander cells, such as lung fibroblasts, were affected by the lack of ERG in endothelial cells, exhibiting increased expression of myofibroblast markers. These findings suggest the pulmonary endothelium may be key to maintain fibroblasts in a quiescent state, corroborating our previous findings (12).

We did not assess ERG deletion at 90 days following tamoxifen injection as whole lungs were used to carry out the scRNA-seq analysis. However, we have assessed ERG deletion at 30 and 45 days (reported in Fig 6A and 7E), and day 60 (unpublished data), post tamoxifen administration and successfully demonstrated ERG deletion at these time points. In addition, the lungs harvested during these time points manifested several vascular abnormalities that we have now reported in Figures 6 and 7. Thus, given that our scRNA-seq data distinctly showed both cellular and transcriptional aberrations, including loss of gCap EC and activation of pro-fibrotic genes in bystander cells, we are confident that ERG would still be largely deleted at day 90 or the manifestations of its deletion still be present.

Reviewer #2 (Remarks to the Author):

This manuscript, written by Caporarello, describes the involvement of a transcription ERG in aging-related lung repair and potential pathogenesis of persistent lung fibrosis, such as idiopathic pulmonary fibrosis (IPF). The authors first compared the epigenomic patterns in lung endothelial cells (ECs) between young and aged mice using ATAC seq. They found that the aged mice generally show a close epigenomic

pattern. They identified binding sites of a transcription factor, ERG, were resided in those differentially regulated regions. They further examined the epigenomic changes in response to a bleomycin stimulation, which is a model of lung injury. They found that ERG may play the pivotal role in the regulation of a series of genes which are involved in lung capillary homeostasis and repair following the injury. Using an in vitro experiment in a human lung microvascular endothelial cells (HLMECs), they conducted siRNA and recapitulated the enhanced fibroblast activation and collagen deposition which they observed from the mouse analyses. They also conducted a single cell RNA sequencing (scRNA seq) analysis of ERG deficient mice. They found that the population of general capillary (gCap) ECs. Finally, they utilized public scRNA seq data of human IPFs and found that the reduced numbers of gCap ECs and vascular progenitor cells are also represented there. Overall, I fully appreciate the authors efforts to deepen our understandings for the relation between aging and etiology of IPF. However, I have substantial remaining concerns as to whether the presented data collectively lead the same conclusion. Even though the collected material should potentially include all the necessary information, some key analyses are missing for every part.

Major points:

1. Given the overall population changes in lung ECs was identified from the scRNA seq analysis, it is essential to further characterize which cell types are responsible for the observed epigenome changes. The observed gene expression changes should be also re-considered, perhaps by making use of a deconvolution analysis. Recently, an analytical method for the simultaneous single cell analysis of ATAC seq and RNA seq (such as scMultiome) has been commercialized. Such an analysis would give an essential information to re-evaluate the results of Figs 1-4.

Response: Our scRNA-seq analysis has revealed gCap ECs as the most affected EC population in the lung lacking endothelial ERG. An in-depth analysis of our bulk RNAseq dataset (based on endothelial cell marker expression) has revealed that the large majority of lung ECs isolated using our lung dissociation method are derived from capillaries (aCap ECs and gCap ECs) (shown in the heatmap below).

Furthermore, our scRNA-seq analysis confirmed the enrichment of lung capillary ECs (Figure 8B-D and F) using our isolation method. Phenotypically, injured lungs from aged mice as well as those from ERG CKO mice exhibited reduced capillary density, suggesting that capillary ECs are more susceptible to aging and to the loss of ERG compared to those derived from large vessels. These

analyses strongly suggest that, while aging-associated epigenetic alterations may broadly affect ECs from different vascular beds, the chromatin alterations we observed using our ATAC-seq analysis likely reflect epigenetic changes that occur in lung capillary ECs. In fact, key gCap EC marker genes, including *Aplnr*, *Tek*, and *Kit* exhibited reduced chromatin accessibility and expression in aged lung ECs relative to young ones (Figure 3F, H, I). We have performed a comparative analysis between the bulk RNA-seq (Aged vs Young) and the scRNA-seq (ERG CKO vs WT), which is now included in the updated Figure 8G. Interestingly, this comparison revealed that, among the genes that were downregulated in both aged ECs and ERG KO ECs, were many gCap EC markers, thus reinforcing the concept that capillaries, especially gCap ECs, are the most affected cells during aging and after ERG deletion. Also, given that gCap ECs exhibit stemness and regenerative properties (13), and that aging greatly affect stem cell regenerative capacity (14), it seems plausible to speculate that abnormal chromatin remodeling in gCap ECs with aging would affect lung repair and fibrosis.

2. Analyses for the scRNA seq data of human IPF is too shallow. The consistency and the inconsistency against the mouse model should be more intensively examined. A particular focus should be put on the status of ERG activity, depending on the patients. Also, I wonder if there are any clinical symptoms associated with the molecular status of the ECs of the patients?

Response: Our study is mainly based on mouse data. However, similarly to IPF lungs, aged mouse lungs, as well as ERG CKO lungs, exhibited vessel rarefaction following injury. Thus, we sought to determine whether IPF lung capillary ECs, more specifically gCap EC, were also compromised compared to endothelial cells from other lung vascular beds. Our scRNA-seq analysis revealed that, similarly to mouse lungs lacking ERG, as well as fibrotic aged mouse lungs, gCap ECs were reduced in IPF lung compared to healthy lungs (Fig.8H-J) and exhibited strong reduction of distinctive gene markers (Fig.8L). In addition, while we did not observe changes in the aCap ECs (aerocytes) in ERG CKO mouse lungs compared to WT, scRNA-seq from IPF lungs showed strong reduction of aCap ECs (Fig.8H-J), suggesting the aCap EC homeostasis is also compromised in end-stage disease lungs. In the revised version of the manuscript, we have now included new observations from the scRNA-seq on IPF lungs. We reported that while IPF lungs showed reduction of both gCap and aCap EC, ECs from other lung vascular beds, including those derived from the systemic and the pulmonary venous circulation, were elevated. While still preliminary, our data shed some new light on the uniqueness of human lung capillary ECs in IPF, their aberrant transcriptional programs, and their potential impact on disease progression.

3. Given that differentially accessible sites in promoters are relatively rare, contribution of ERG for the overall etiology of IPF should be carefully considered.

Response: Although differentially accessible sites in promoters are relatively rare (based on our analysis), we were still able to identify multiple chromatin alterations in promoters especially in those genes implicated in EC identity and angiogenesis. While the analysis of the human IPF scRNA-seq data is still very preliminary, we did observe that, similarly to mouse lungs lacking ERG, human IPF lungs exhibited reduced number of gCap ECs compared to healthy lungs (Fig.8H-J) as

well as distinctive gCap EC markers (Fig.8L). In addition, scRNA-seq analysis of WT and ERG CKO animals revealed that neutrophils, macrophages, and fibroblasts of ERG CKO mice exhibit increased expression of genes encoding for fibrogenic mediators that have been previously implicated in IPF (new data now included in the Supplemental Figure 7), further confirming that dysfunctional ERG signaling in the aged lung and during persistent fibrosis may enhance pathogenic responses in bystander lung cells.

Minor points:

4. I wonder when the function of ERG is deteriorated with aging to eventually give an epigenomic condition which is vulnerable to IPF. Increasing data points would answer the question.

Response: We agree that identifying when the function of ERG deteriorates with aging would complement our study. However, this type of analysis is very challenging. Given that the ERG signature we identified in aged EC is based on differentially accessible chromatin regions and not on ERG activity/function, and that aging-associated chromatin alterations may directly limit ERG chromatin recruitment, it is very difficult to assess when during aging these chromatin alterations may occur. While interesting, we feel that this investigation is beyond the scope of this work.

5. When and how does ERG realize the most important role after the injury. Further detailed data collection and analyses should be needed before during the periods of inflammatory responses, lung vascular repair and the establishment of eventual fibrosis. For the fibrosis, the data points are too limited to discuss the “persistence”.

Response: Our bleomycin study on ERG CKO mice demonstrated that ERG deletion after the peak of fibrosis (day 14) led to delayed fibrosis resolution. In addition, we found that the expression of numerous ERG target genes was elevated in young ECs during the initial resolution phase of fibrosis (day 30 post bleomycin). While focused on this particular time point, our study demonstrates that endothelial ERG plays a critical role during the resolution phase of lung fibrosis following injury, and that its impaired function may lead to persistent fibrosis. However, to address this reviewer’s concern, we have edited the text of the revised manuscript and changed “persistent fibrosis” with “impaired fibrosis resolution” in young mouse lungs in absence of endothelial ERG.

6. The authors have already identified *Nos3* as a key player for persistent lung fibrosis (reference 17). I wonder *Nos3* is the major target of ERG? In other words, the function of ERG is mostly in proper induction of *Nos3*? Further careful analyses should be needed for additional possible roles of ERG.

Response: This is a very interesting point. Previous studies showed that ERG binds to the promoter of *NOS3* gene to regulate its expression (11). Furthermore, a recent paper showed that *NOS3* expression is reduced in the lung of ERG deficient mice (15), and our laboratory demonstrated that ERG silencing in human lung capillary ECs inhibited *NOS3* gene expression (data not shown). In addition, we previously reported a critical role for eNOS during lung fibrosis resolution in young mice. Given that *Nos3*, along with other ERG target genes, was upregulated during lung fibrosis resolution in young mice, but not in aged mice with persistent fibrosis, we think that dysfunctional ERG/*NOS3* signaling during aging may be, at least in part, responsible for the adverse outcomes.

However, besides *NOS3*, we found numerous ERG target genes associated with vascular repair, including *Bmp2*, *Lama4*, and *Col4a1*, whose expression was increased in young EC but not in aged ECs, thus reinforcing the concept that ERG regulates multiple signaling pathways in ECs that, when altered, may result in lung vascular disrepair and delayed fibrosis resolution.

7. Degree of differential accessibility and the gene expression levels should be mutually compared.

Response: We have included in the updated Figure 3G and 4I and J the correlation analysis between ATAC-seq and RNA-seq.

8. Fig. 6: Essential controls seem missing from ERG CKO (Bleomycin minus). This control is important because it is possible that ERG CKO may have developed predisposed lesions.

Response: We have included in our revised version a new figure (Figure 6) containing a comprehensive characterization of ERG CKO lungs in the absence of bleomycin. Besides vascular leak, which was reported in the original version of our manuscript, we have now carried out hydroxyproline measurements and trichrome staining of WT and ERG CKO lungs to assess lung remodeling and fibrosis. In addition, we have also included a new FACS analysis showing increased lung neutrophils in the lung of ERG CKO mice. These changes were also accompanied by increased lung inflammation as previously reported (2).

9. Figure 6C: The timing for the conditional KO of ERG is not clear to me. If this is a model of an aged lung, the KO should be induced before the Bleomycin treatment?

Response: Our multi-omics analysis showed an ERG signature in lung ECs during the initial resolution phase of post bleomycin induced lung fibrosis (day 30). Thus, in order to assess the specific role of ERG during lung fibrosis resolution we began tamoxifen treatment during the peak of inflammation to ensure complete deletion of ERG by day 30. However, given that endothelial ERG deletion in absence of injury leads to pulmonary vascular leak, vascular remodeling, and inflammation, we expect that deleting ERG prior to bleomycin treatment would result in a more severe lung inflammation and fibrosis (as observed in aged mice).

10. The first sentence of the abstract is misleading. Strictly speaking, I do not think this is the paper on human IPF.

Response: We have edited the abstract, as suggested.

11. Discussion section is, at least to some extent, repeating the descriptions in the former sections, thus, could be shortened.

Response: we have shortened the discussion section.

Reviewer #3 (Remarks to the Author):

This manuscript from Ligresti and colleagues reported that dysregulation of ERG impairs lung vascular repair and promotes fibrosis in aging mice. Although the proposed data would be interesting, the study lacks a logical connection and clear mechanistic insights. The data analysis should be improved and more experimental data are required for demonstrate the mechanism about how ERG mediates vascular repair in the aging mice.

Here are some important concerns to highlight:

1. The sequencing data analysis should be improved. The data in Fig.1A does not support the authors' statement that "extensive differences in chromatin accessibility, with samples clustering into different groups based on age and responses to bleomycin injury". In fact, the Sham and Bleo group mixed together for the 3 of 5 replicates in the aged mice, representing the data are not so qualified.

Response: We agree with the reviewer and changed our original statement to better reflect the transcriptional responses of aged lung ECs to bleomycin. As stated above (Rev#1), we observed inherent inter-individual heterogeneity in chromatin accessibility associated with aging. In fact, while young ECs (sham and injured) clustered quite uniformly in our PCA analysis study (Fig. 1A), aged ECs, exhibited a more scattered distribution, suggesting that aged mouse ECs produce different responses to bleomycin likely due to their underlying more heterogeneous chromatin accessibility.

2. For Fig.1B, the author cannot just use squares for regions with divergent chromatin accessibility, which is quite inaccurate and unacceptable. The authors should divide these regions into different groups according to the accessibility change patterns and then plot them separately.

Response: We have now updated the Figure 1B and removed the squares from the heatmaps. To highlight the regions with different chromatin accessibility, we split the relative number of peaks for each comparison in 3 plots displaying 1) regions of reduced chromatin accessibility; 2) increased chromatin accessibility; 3) unchanged chromatin accessibility. This new analysis is now included in the Supplementary Figure 2.

2. The authors have performed ATAC-seq and RNA-seq, but analyzed the data separately, it would be better to perform joint analysis combined ATAC-seq with RNA-seq. For ATAC-seq, taking Fig.1F as example, the author need to check if those more/less accessible loci are relative to their proximal gene up/down regulation in RNA-seq. Besides, it would be better to show chromatin accessibility and gene expression with the genomic plot by IGV etc instead of only the gene names in case of any misinterpretation because ATAC-seq peak are mostly located at distal regulatory elements such as enhancers but not gene, and each gene would include several ATAC-seq peaks.

Response: We performed a correlative analysis between ATAC-seq and RNA-seq and the findings are now shown in the updated Figures 3G and 4I and J. We have removed the gene annotation in figure 1F and included both genomic snapshots and gene expression of ERG-regulated genes in

figure 3H and I and 4K-M. Given that locating enhancers is very challenging using ATAC-seq, only differentially accessible promoters were included in the comparative analysis between RNA-seq and ATAC-seq analysis.

4. For Fig.2, it is really difficult to follow the logic of such analysis. It would be better if the authors can perform the comparative analysis that how many sites that are accessible in the young mice but closed in the old mice, and also take the reverse part (open in the old mice but closed in the young). Similarly, loci groups relative to Bleo response can be summarized and then motif enrichment analysis should be performed for those sub-clusters of ATAC-seq peaks.

Response: Because this is a bulk ATAC-seq analysis, the genomic peaks that we have identified are the results of an average of “Open” and “Close” chromatin states. Thus, we are not able to accurately determine “Open” or “Close” chromatin regions but rather less or more accessible chromatin regions relative to a different condition. For example, for the comparison Young ECs vs Aged ECs (Uninjured) we have first determined the chromatin regions that were differentially accessible in aged ECs relative to young ECs. We then found that among the differentially accessible chromatin regions, roughly 80% were less accessible and 20% more accessible in aged ECs relative to young ECs (the other way around would be 80% more accessible and 20% less accessible in young ECs relative to aged lung ECs). Thus, our motif analysis was performed on this sub-cluster of differentially accessible chromatin regions.

5. The ERG ChIP-seq data are required for the validation of the potential target genes of Fig. 2J-L.

Response: ChIP sequencing on freshly isolated ECs is very challenging due to the limited number of cells that we can harvest from mouse lungs. To address this concern, we have interrogated a publicly available ChIP-seq from ERG-silenced HUVECs (7). This study was designed to investigate the contribution of endothelial ERG to histone mark decoration, specifically the deposition of H3K27ac mark (active transcription). Intriguingly, we found that numerous endothelial genes whose H3K27ac mark deposition and transcription were impaired in ERG-silenced ECs *in vitro* exhibited reduced chromatin accessibility with aging (Figure 2J-L), suggesting that dysfunctional ERG signaling with aging may impair histone decoration and gene transcription. As an example, below are three genomic snapshots showing reduced H3K27ac marks in the promoters of APLNR, KIT and PLVAP genes in ERG silenced ECs. The expression of these genes was found to be reduced with aging.

6. Again, the Fig3A does not support the authors' claim that "PCA analysis showed that samples formed distinct clusters, with ECs of injured and uninjured lungs sharing the least similarities", the samples can't be sperate correctly by both PC1 and PC2, and showing that significant difference between replicates, which also simply reflected by Fig.3D-F. We have some concerns about the data quality and consistency.

Response: Given that our multi-omics analysis was carried out on freshly isolated lung ECs, the amount of RNA yielded is quite low. Thus, we used a low input mRNA preparation library, which may increase variability between samples. We agree with the reviewer and we have edited the text to better reflect our findings. In addition, as detailed above (Reviewer #1), all samples had a good quality RNA. We have included in the revised manuscript the RNA integrity number (RIN) for each sample.

7. For Fig.3B-C, how did the author perform that analysis? Detailed statement about the analysis pipeline and the corresponding citations should be included in the method part.

Response: we have updated the method section with details on IPA analysis and added corresponding citations in the references section, as suggested.

8. Fig.3H, the KD efficiency need to be shown.

Response: We have included in the updated Fig. 3J the validation of ERG silencing in human EC *in vitro*.

9. Line 248 & Line 394, the data need to be shown.

Response: We have now included these data in the manuscript (Supplementary Figures 3 and 6) as suggested and edited the text accordingly.

10. Fig.7B-D, the color scheme makes the figure difficult to read. Moreover, the expression of all the marker genes for each cell type should be shown in the supplementary figures.

Response: We have changed the color scheme of Figure 7B-D to improve clarity. We have also included representative gene markers for each cell type in the supplementary material (Supplementary Fig.5)

11. Lack of the molecular mechanism makes the manuscript incomplete, it is still unclear why ERG are important for capillary homeostasis, and it is also unclear whether the gCap cells contribute to injury repair. An increased focus on ERG coupled mechanisms may improve the manuscript.

Response: While we have not yet investigated the specific molecular mechanisms through which ERG regulates capillary homeostasis, our data suggest that dysfunctional ERG signaling mainly affects gCap ECs. In fact, our multi-omics analysis demonstrated a reduced expression of several gCap EC markers (e.g., *Aplnr*, *Tek*, *Kit* and *Plvap*) in aged lung ECs, as well as in young ECs lacking ERG, suggesting a putative role for ERG in lung capillary homeostasis. Although ERG is broadly expressed in lung ECs, our data suggest that ERG function may vary in ECs from different vascular beds. Future studies will be conducted in our laboratory to shed further light into the role

of ERG and its downstream molecular effectors in lung capillary homeostasis vs other types of lung ECs. In regard to the contribution of the gCap EC to vascular repair, a recent article has reported a key role for gCap ECs in vascular repair (13). Specifically, gCap ECs serve as specialized stem/progenitor cells, orchestrating lung capillary regeneration following injury.

Minor issues:

1. For Fig.1C, the “open/close” should be “more/less accessible” for consistence and to avoid misleading.

Response: We agree and changed the nomenclature in Fig.1

2. LINE153, Homer need a citation.

Response: We have included homer citation in the bibliography.

Reviewer #4 (Remarks to the Author):

The manuscript entitled “Dysregulation of the transcription factor ERG in aging impairs lung vascular repair and promotes persistent fibrosis” by Caporarello and collaborators describe the involvement of ERG signaling in the homeostasis and response to repair of lung vasculature. The topic of manuscript is exiting and highly relevant. However, some conceptual and experimental designs raise major concerns that limit my enthusiasm for this manuscript.

Major concerns:

1. The authors open with this statement: “Here, we employed epigenetic and transcriptional analysis using ATAC-seq and RNA-seq of freshly isolated lung endothelial cells (ECs) from young and aged mice to identify endothelial programs during the resolution or progression of bleomycin-induced lung fibrosis.” However, the author used 30 days post bleomycin instillation as their main endpoint. In their previous publication (2020 Aging Cell -figure1c-d), the levels of hydroxyproline in the lung of the young mice at day 30 is as high (if not higher) than the older mice. Only at day 75 post-installation they report a difference. That will be the correct time window to address any changes in resolution, since (base on their own data) day 30 is still a highly fibrotic. Further, the authors should compare cell populations and gene expression in endothelial cells from resolved fibrosis to peak of fibrosis (i.e. both animals subjected to bleomycin application).

Response: We chose the 30-day time point based on the transcriptional responses we observed in lung fibroblasts from young mice. We recently demonstrated that while lung hydroxyproline levels were still elevated at day 30 post bleomycin injury in young mice, lung fibroblasts in these mice were transcriptional more quiescent compared to those isolated during the peak of fibrosis (Day 14) (16). For example, we demonstrated that *Col1a1* (encoding for Collagen-1), along with other pro-fibrotic genes, was strongly induced in lung fibroblasts at day 14 post bleomycin challenge,

however, its expression level dropped to baseline at day 30 post bleomycin. If we had chosen day 75 as the time point to assess transcriptional changes in endothelial cells, we could have missed important transcriptional responses that may have occurred during the initial phase of lung fibrosis resolution.

2. ERG cKO studies (Figure 6): The design of this animal experiments is lacking the appropriate controls. Tamoxifen has some anti-angiogenic effects, so the proper control should be Erg fl/fl mice with tamoxifen and PBS instillation (instead of only PBS). This would be a more comparable uninjured WT model. In addition, the authors do not report any fibrotic-related data on the changes (or absence of changes) in the ERG cKO when uninjured. Is the vascular leakage promoting fibrotic changes in the uninjured cKO mice? Also, is this leakage affecting different size vessels differently?

Response: We have now included the control Erg^{flox/flox} with tamoxifen and PBS instillation to the experiment, as suggested (new Figure 7B and C). In addition, we have included a new figure (Figure 6) reporting hydroxyproline content and trichrome staining of WT and ERG CKO lungs in the absence of injury (Figure 6 F and G). While hydroxyproline content did not change between WT and ERG CKO lungs, trichrome staining revealed abnormal vascular remodeling around large vessels as well as small alveolar capillaries. We have also included a FACS analysis (Fig. 6C) demonstrating elevated lung neutrophils in the lung of ERG CKO compared to WT lungs. Altogether these new data suggest that abnormal perivascular signals and increased lung inflammation contribute to the unresolved fibrosis observed in ERG CKO mice upon bleomycin challenge.

3. Figure 7: In this study a third different timepoint (90 days after recombination) is used. None of these data then show changes than can be used to complement Figure 6. In addition, at this point of the manuscript the authors bring up the different sources of the ECs in the lung (all previous studies were done as if there were one single population). If the authors want to venture in the understand the changes in aCap and gCap with aging (and in the context of the ERG KO), this single cell study should be compared with aged mice (both WT and KO).

Response: We used different time points because we wanted to highlight different functions of ERG as follows: 1) ERG contribution to lung fibrosis resolution in young mice (which spontaneously resolves at 45 days after bleomycin); 2) Assess whether ERG loss in ECs recapitulates spontaneous fibrotic features in the absence of injury, for which we reasoned a longer time point (90 days) is needed.

To shed further light on the contribution of ERG to vascular aging, we have compared our bulk RNA-seq (Aged vs Young) with the scRNA-seq (ERG CKO vs WT) and generated new data which are now included in the revised version of the manuscript. Intriguingly, among the differentially expressed genes shared by the 2 datasets, gCap ECs markers (e.g., *Aplnr*, *Kit*, *Tek*, *Sema3g*, *Efn1*) were downregulated (Figure 8G).

4. Methods: In the description of the mouse model, detailed information of the type of anesthesia and concentration/regimen of the tamoxifen injection should be added. Equally important should be to describe

in detail the experimental set-up of the cell culture studies (i.e. concentration of siRNA, time of transfection, time of harvest, etc). These additions will ensure the reproducibility of any of the experiments.

Response: We added this information into the method section, as suggested.

Minor concerns

Figure 1: Despite the labeling, it is not evident that panel A & B represent data from the four experimental groups and the subsequent panel are just the 3 comparisons between groups. Seems confusing upon first reading and limits the excitement of the data.

Response: In Figure 1A and B, we show the overall changes in chromatin accessibility for each condition. The subsequent panel C shows the comparison between young bleo and aged bleo relative to young uninjured.

Figure 3: The use of blue as a group identificatory “aged sham” on panel A plus as part of the colored scale in the heatmap is confusing. Also, color scheme used in panel G is not carried equally for all the TSS analysis.

Response: We have updated the figure, as suggested.

Figure 4: The color scheme is not consistent between the same groups on panels I, J and K.

Response: It is not always possible to change the appearance of the genomic snapshots as the ATAC peak colors are randomly assigned to each gene by the IGV software.

Figure 5: Microscopy images in this figure are lacking proper scale bars. In addition, panel H does not add any results or information not already mentioned in the method section. Similarly, the schemes in panels A and D seems superfluous and are missing key information such as harvesting time or dilution ration of the conditioned media in fresh media (if any).

Response: The updated version of the figures includes now scale bars. We have also updated the schematics and incorporated experimental details.

Figure 6: Except for panel A, all other microscopy images in this figure are lacking proper scale bars.

Response: We have included appropriate scale bars, as suggested.

Methods: catalog numbers for the commercially available mice are missing. In addition, clone identification (or catalog number) should be added to the antibodies using in this study.

Response: The revised version of the manuscript includes the catalog numbers for the commercially available mice. We have also included catalog numbers for the antibodies in the method section of the manuscript.

References

1. Angelidis I, Simon LM, Fernandez IE, Strunz M, Mayr CH, Greiffo FR, et al. An atlas of the aging lung mapped by single cell transcriptomics and deep tissue proteomics. *Nat Commun*. 2019;10(1):963.
2. Sperone A, Dryden NH, Birdsey GM, Madden L, Johns M, Evans PC, et al. The transcription factor Erg inhibits vascular inflammation by repressing NF-kappaB activation and proinflammatory gene expression in endothelial cells. *Arterioscler Thromb Vasc Biol*. 2011;31(1):142-50.
3. Yuan L, Nikolova-Krstevski V, Zhan Y, Kondo M, Bhasin M, Varghese L, et al. Antiinflammatory effects of the ETS factor ERG in endothelial cells are mediated through transcriptional repression of the interleukin-8 gene. *Circ Res*. 2009;104(9):1049-57.
4. Looney AP, Han R, Stawski L, Marden G, Iwamoto M, and Trojanowska M. Synergistic Role of Endothelial ERG and FLI1 in Mediating Pulmonary Vascular Homeostasis. *Am J Respir Cell Mol Biol*. 2017;57(1):121-31.
5. Hogan NT, Whalen MB, Stolze LK, Hadeli NK, Lam MT, Springstead JR, et al. Transcriptional networks specifying homeostatic and inflammatory programs of gene expression in human aortic endothelial cells. *Elife*. 2017;6.
6. Dryden NH, Sperone A, Martin-Almedina S, Hannah RL, Birdsey GM, Khan ST, et al. The transcription factor Erg controls endothelial cell quiescence by repressing activity of nuclear factor (NF)-kappaB p65. *J Biol Chem*. 2012;287(15):12331-42.
7. Kalna V, Yang Y, Peghaire CR, Frudd K, Hannah R, Shah AV, et al. The Transcription Factor ERG Regulates Super-Enhancers Associated With an Endothelial-Specific Gene Expression Program. *Circ Res*. 2019;124(9):1337-49.
8. Shah AV, Birdsey GM, Peghaire C, Pitulescu ME, Dufton NP, Yang Y, et al. The endothelial transcription factor ERG mediates Angiopoietin-1-dependent control of Notch signalling and vascular stability. *Nat Commun*. 2017;8:16002.
9. Uehara M, Enomoto N, Mikamo M, Oyama Y, Kono M, Fujisawa T, et al. Impact of angiopoietin-1 and -2 on clinical course of idiopathic pulmonary fibrosis. *Respir Med*. 2016;114:18-26.
10. Barkaway A, Rolas L, Joulia R, Bodkin J, Lenn T, Owen-Woods C, et al. Age-related changes in the local milieu of inflamed tissues cause aberrant neutrophil trafficking and subsequent remote organ damage. *Immunity*. 2021;54(7):1494-510 e7.
11. Shah AV, Birdsey GM, and Randi AM. Regulation of endothelial homeostasis, vascular development and angiogenesis by the transcription factor ERG. *Vascul Pharmacol*. 2016;86:3-13.
12. Caporarello N, Meridew JA, Aravamudhan A, Jones DL, Austin SA, Pham TX, et al. Vascular dysfunction in aged mice contributes to persistent lung fibrosis. *Aging Cell*. 2020:e13196.
13. Gillich A, Zhang F, Farmer CG, Travaglini KJ, Tan SY, Gu M, et al. Capillary cell-type specialization in the alveolus. *Nature*. 2020;586(7831):785-9.
14. Sameri S, Samadi P, Dehghan R, Salem E, Fayazi N, and Amini R. Stem Cell Aging in Lifespan and Disease: A State-of-the-Art Review. *Curr Stem Cell Res Ther*. 2020;15(4):362-78.
15. Peghaire C, Dufton NP, Lang M, Salles C, II, Ahnstrom J, Kalna V, et al. The transcription factor ERG regulates a low shear stress-induced anti-thrombotic pathway in the microvasculature. *Nat Commun*. 2019;10(1):5014.
16. Tan Q, Link PA, Meridew JA, Pham TX, Caporarello N, Ligresti G, et al. Spontaneous Lung Fibrosis Resolution Reveals Novel Antifibrotic Regulators. *Am J Respir Cell Mol Biol*. 2021;64(4):453-64.

Reviewers' Comments:

Reviewer #1:

Remarks to the Author:

The Authors have addressed the main concerns and the Manuscript has improved significantly

Reviewer #2:

Remarks to the Author:

First of all, I appreciate the substantial efforts of the authors to revise this manuscript. With a series of extensive analyses and deepened discussion, I think this manuscript has been very much improved. Although I still have a remaining concern. That is, the association between the transcriptomic and epigenomic features should be examined in a more direct manner, given those two layers appeared to have a mutually unique features. At least potentially, the target tissue or, even after the separated cells, still consist of heterogeneous cell types. Therefore, it is possible that additional single cell "epigenome" analysis may lead to a no less important insight than that focusing on the capillary ECs, which might have been overlooked by scRNA seq analysis. Nevertheless, I have come to consider that those issues should be addressed in their future studies. The contents of the manuscript are already rich and further expanding the contents would make this paper difficult to understand for non-expert readers. Indeed, I sincerely hope that the authors continue their efforts to elucidate the etiology of this disease in its association with aging.

Reviewer #3:

Remarks to the Author:

I appreciate the authors' efforts to improve the joint analysis of ATAC-seq and RNA-seq data. However, and all the ATAC-seq analyses were focused on the promoters, while dynamic ATAC-seq peaks are mostly located at distal regulatory elements such as enhancers but not gene promoters according to our experience, and also proved by the authors' data (Fig.1E), which may lead to misinterpretation of the mechanism of the whole study.

For the previous question 5, the authors did not reply to my question, published ERG ChIP-seq data in HUVECs are available and only 3 genes are not representable, the statistics results need to be shown.

Overall, the phenomenons reported by the authors are interesting, but the biased analysis pipeline and lacks the experimental mechanism validation limit my enthusiasm for this paper.

Reviewer #5:

Remarks to the Author:

The authors have sufficiently addressed all of the concerns of the 4th reviewer and i have no additional concerns.

Response to reviewers: Manuscript NCOMMS-21-17167A

We thank the reviewers for their effort in evaluating our manuscript, their constructive comments, and the recognition of our work

Reviewers' comments

Reviewer #1 (Remarks to the Author):

The Authors have addressed the main concerns and the Manuscript has improved significantly.

Response: We thank the reviewer for recognizing our effort in improving the revised manuscript.

Reviewer #2 (Remarks to the Author):

First of all, I appreciate the substantial efforts of the authors to revise this manuscript. With a series of extensive analyses and deepened discussion, I think this manuscript has been very much improved. Although I still have a remaining concern. That is, the association between the transcriptomic and epigenomic features should be examined in a more direct manner, given those two layers appeared to have a mutually unique feature. At least potentially, the target tissue or, even after the separated cells, still consist of heterogeneous cell types. Therefore, it is possible that additional single cell "epigenome" analysis may lead to a no less important insight than that focusing on the capillary ECs, which might have been overlooked by scRNA seq analysis. Nevertheless, I have come to consider that those issues should be addressed in their future studies. The contents of the manuscript are already rich and further expanding the contents would make this paper difficult to understand for non-expert readers. Indeed, I sincerely hope that the authors continue their efforts to elucidate the etiology of this disease in its association with aging.

Response: We thank reviewer for the gratifying and encouraging comments. We agree with the reviewer's comments regarding the heterogenic nature of the isolated lung endothelial cells. Indeed, we also think that epigenetic alterations associated with aging or lung injury may differentially affect lung EC sub-populations. We recognize that besides aged capillary ECs, which are the focus of our manuscript, other EC populations may contribute to aging-associated dysfunctional responses. Therefore, as suggested by the reviewer, we are planning to carry out future epigenetic analysis to further elucidate endothelial chromatin abnormalities associated with aging at single cell resolution.

Reviewer #3 (Remarks to the Author):

I appreciate the authors' efforts to improve the joint analysis of ATAC-seq and RNA-seq data. However, and all the ATAC-seq analyses were focused on the promoters, while dynamic ATAC-seq peaks are mostly located at distal regulatory elements such as enhancers but not gene promoters according to our experience, and also proved by the authors' data (Fig.1E), which may lead to misinterpretation of the mechanism of the whole study.

Response: We thank the reviewer for appreciating our effort in improving our revised manuscript. As for our choice for selecting ATACseq peaks within promoters (+/- 1kb of the TSS), we agree with the reviewer about the importance of distal regulatory elements, such as enhancers, in coordinating transcriptional responses to lung injury and in aging. In fact, most of our study takes in consideration all potential regulatory elements found within ATAC peaks to understand global changes in chromatin accessibility. However, given the current state of mouse regulatory element annotation repositories, we are limited to cell-specificity and predicted position without sufficient data to accurately identify gene targets of these regulatory elements. Although there are tools available to explore these dynamics, such as GeneHancer for human and GREAT for human and mouse, the majority of peaks are presented with multiple potential target genes resulting in the incorporation of potential false-positive associations. As a result, we decided not to incorporate distal regulatory regions for the ATAC-seq/RNA-seq comparative analysis. We decided to focus on regions that are most likely to directly impact the expression of nearby genes. Using this approach, however, we were still able to identify a large number of aging-associated chromatin alterations in ERG regulated genes implicated in endothelial cell identity and angiogenesis.

For the previous question 5, the authors did not reply to my question, published ERG ChIP-seq data in HUVECs are available and only 3 genes are not representable, the statistics results need to be shown. Overall, the phenomenons reported by the authors are interesting, but the biased analysis pipeline and lacks the experimental mechanism validation limit my enthusiasm for this paper.

Response: We decided to interrogate publicly available ChIP-seq datasets that were generated in HUVECs to validate our findings in silico. We fully acknowledge the limitation of this approach; however, this was the only published work that investigated ERG contribution to chromatin remodeling in vascular endothelial cells. Because of the limited number of samples (one sample per condition), we were unable to carry out a meaningful statistical analysis.

Although in the previous revision we have mentioned about the limitations that are associated with ChIP sequencing using freshly isolated lung ECs, we recognize that we did not provide the reviewer sufficient details concerning the technical, temporal, and financial limitations that prevent us from carrying out this analysis. First, while ChIP sequencing technology has advanced significantly over the past years, limitations still remain, especially regarding the number of cells needed to achieve reliable results. We have consulted with experts in our epigenomic core facility, and they said that whereas the number of cells needed for histone mark pull down using low input ChIP-seq protocol can be relatively low, for transcription factor pull down they recommend approximately 1-2 million of cells using a low input ChIP protocol. While we can obtain this cell number using cultured cells, in the *in vivo* setting the number of ECs that we are able to isolate from a single mouse lung is limited (approximately 200K using our digestion method). Based on these observations, we would need to pool a minimum of 5 mouse lungs per condition to obtain the proper number of cells needed to carry out a single ChIP experiment. Given that we have four experimental conditions run in triplicate, we would need approximately 30 mice to assess ERG binding in young and aged lung ECs, and approximately 60 mice to extend the aging assessment to the bleo-treated groups. While we agree that this would be an interesting experiment, we also feel that this additional analysis would require substantial time commitment and a large financial investment.

As an alternative approach, we attempted to computationally assess ERG binding at transcription factor binding sites (TFBS) using ATAC foot-printing. Although our ATAC dataset met the current ENCODE standards of greater than 15 million fragments for the analysis of differentially accessible chromatin regions (<https://www.encodeproject.org/data-standards/atac-seq/atac-encode4/>), the depth on a *per sample* basis was not sufficient to perform sample specific footprinting analysis. As a result, we combined our samples based upon age of “Young vs. Aged” to surmount roughly 150 and 120 million reads per age group, respectively (the minimum recommended is 200 million reads). Despite observing a potential footprint identity of ERG in our visualization as well as differences in occupancy between different conditions, we think that the level of coverage for both the flanking regions and our potentially bound region was not sufficient to confidently use these data to support our original claims. We believe that even with the concatenation of our samples we will not be able to achieve sufficient read depth coverage. As a result, we do not wish to include the ATAC-seq foot-printing analysis in our revised manuscript.

Whereas we recognize that our work still requires further investigations to fully understand the mechanisms regulating endothelial ERG chromatin interaction in aging and in response to lung injury, we also feel that our data conclusively demonstrate for the first time that endothelial ERG plays a critical role during lung repair and fibrosis, and that chromatin loci containing ERG binding motifs were differentially accessible in young vs aged lung ECs and fibrotic aged lungs. While we acknowledge the limitation of the ATAC-seq approach, especially in regard to the investigation of transcription factor/chromatin binding, this epigenetic analysis was fundamental to identify ERG as a putative transcription factor implicated in the aberrant transcriptional response of aged lung EC. Our study also provides strong evidence that loss of ERG during the initial resolution phase following lung injury recapitulates the phenotypic manifestations observed in fibrotic aged lungs. In addition, a comparative analysis between our scRNA-seq on ERG CKO lungs and the bulk RNA-seq from isolated aged lung ECs demonstrated that numerous capillary specific gene markers exhibited reduced expression in endothelial cells lacking ERG as seen in aged lung ECs compared to young ones.

Reviewer #5 (Remarks to the Author):

The authors have sufficiently addressed all of the concerns of the 4th reviewer and i have no additional concerns.

We thank the Reviewer for this comment.